# *Operando* monitoring of thermal runaway in commercial lithium-ion cells via advanced lab-on-fiber technologies

Wenxin Mei [1,6], Zhi Liu[2,6], Chengdong Wang[1], Chuang Wu[2], Yubin Liu[2], Pengjie Liu[1], Xudong Xia[2], Xiaobin Xue[2], Xile Han[2], Jinhua Sun[1], Gaozhi Xiao[3], Hwa-yaw Tam[4], Jacques Albert[5], Qingsong Wang [1] ✉ & Tuan Guo [2] ✉

Operando monitoring of complex physical and chemical activities inside rechargeable lithium-ion batteries during thermal runaway is critical to understanding thermal runaway mechanisms and giving early warning of safety-related failure. However, most existing sensors cannot survive during such extremely hazardous thermal runaway processes (temperature up to 500 °C accompanied by fire and explosion). To address this, we develop a compact and multifunctional optical fiber sensor (12 mm in length and 125 μm in diameter) capable of insertion into commercial 18650 cells to continuously monitor internal temperature and pressure effects during cell thermal runaway. We observe a stable and reproducible correlation between the cell thermal runaway and the optical response. The sensor's signal shows two internal pressure peaks corresponding to safety venting and initiation of thermal runaway. Further analysis reveals that a scalable solution for predicting imminent thermal runaway is the detection of the abrupt turning range of the differential curves of cell temperature and pressure, which corresponds to an internal transformation between the cell reversible and irreversible reactions. By raising an alert even before safety venting, this new operando measurement tool can provide crucial capabilities in cell safety assessment and warning of thermal runaway.

As leading electrochemical energy storage and conversion devices in our daily lives[1,2], lithium-ion batteries have been identified as critical components in the transition from depleted fossil fuels to sustainable worldwide energy production and use. This has not been without issues, however, as recurrent fire safety incidents have become a serious source of concern impacting their further development[3,4]. In this regard, it is essential to improve our understanding of the mechanism of lithium-ion battery thermal runaway and its evolution to inform methods to prevent fires and explosions or at the very least

to provide means for early warnings before catastrophic events occur[5,6].

Lithium-ion battery safety issues originate from thermal runaway within a cell, which is defined as an uncontrolled temperature rise[7] provoked by exothermic chain reactions that often become catastrophes in the form of fire and explosions[8]. There is an urgent need to understand this thermal runaway mechanism in order to reduce its occurrence and develop early warning and prevention measures. The commonly used indicators observed during thermal runaway include

[1]State Key Laboratory of Fire Science, University of Science and Technology of China, Hefei 230026, China. [2]Institute of Photonics Technology, Jinan University, Guangzhou 511443, China. [3]Advanced Electronics and Photonics Research Centre, National Research Council of Canada, Ottawa, ON K1A 0R6, Canada. [4]Department of Electrical Engineering, The Hong Kong Polytechnic University, Kowloon, Hong Kong SAR, China. [5]Department of Electronics, Carleton University, Ottawa, ON K1S 5B6, Canada. [6]These authors contributed equally: Wenxin Mei, Zhi Liu. ✉e-mail: pinew@ustc.edu.cn; tuanguo@jnu.edu.cn

surface temperature and temperature rise rate, voltage drop, venting, mass loss, gas release and fire, which are mostly detected through external electrical, heat, sound, and gas measurements and analysis[9,10]. Although straightforward, these "external" measurements can only supply inadequate and time-delayed information about the internal source of the onset of thermal runaway processes and of its true time evolution. The abrupt generation and accumulation of excessive heat inside a cell is dissipated only with difficulty, resulting in damaging temperature gradients and hysteresis that are undetected by external measurements, thus preventing their thorough analysis and the development of countermeasures and safeguards. While sensors currently exist to monitor the needed parameters in cells[11], for example in situ temperature monitoring by inserting thermocouples into the cell[12], unfortunately most cannot survive during the extremely hazardous thermal runaway process in which internal temperature can increase to 500 ~ 800 °C and the chemical environment is challenging. It is therefore imperative to develop new noninvasive, miniature, robust and remotely operated "in-cell" solutions for thermal runaway mechanisms and their operando evolution process[13] (i.e., while in use and without interfering with cell operation) that enable decoding of the relevant internal parameters with improved temporal and spatial resolution.

Optical fiber sensors represent an ideal solution for these tasks due to their flexibility, small size, light weight, high temperature and pressure survivability, chemical resistance, absence of conductivity and immunity to electromagnetic interference[14–16], while providing superior measurement performance compared to other thermoelectric sensors. Optical fiber sensors operate by correlating some of the many characteristic parameters (such as wavelength, intensity, phase, polarization state, etc.) of the interrogating light signals with factors such as local temperature, stress, pressure, and refractive index, which are all crucial parameters for battery monitoring[17]. Integrating optical fiber sensors inside batteries was first reported by Pinto's group for real-time temperature measurements[18], following with a series of studies for multi-point measurements[19–22]. Subsequently, they further proposed advanced sensing schemes for cells internal strain and temperature discrimination by using a polarization-maintaining Fiber Bragg Grating (FBG)[23] and a hybrid FBG and Fabry–Perot interferometer (FPI)[24]. Recently, to obtain chemical kinetics and thermodynamic information during charge–discharge cycling, a team led by Tarascon's group managed to estimate the solid electrolyte interface (SEI) formation and structural evolution by measuring the cell internal temperature and pressure with FBG sensors, and heat generation by the cell was determined without resorting to microcalorimetry[25]. More recently, another team led by Huang implanted FBG sensors into a lithium–sulfur cell to measure the cathode stress evolution, showing a strong correlation between stress and cathode structure[26]. The on-going efforts and developments of many new and valuable optical fiber sensing approaches, including tilted fiber Bragg grating sensors[27,28], hollow-core optical fiber sensors[29] and infrared fiber evanescent wave spectroscopy[30], that can in situ monitor electrolyte/electrode chemistry were reported in the past few years, dramatically enhancing our understanding of degradation, aging, and prognostic assessment. These examples all demonstrate the feasibility of inserting an optical fiber into a battery and extracting important internal characteristics, during normal charge–discharge cycling of the cell. However, such measurement has not been attempted during battery thermal runaway due to the lack of suitable sensors capable of surviving and decoding the complex physical-chemical-electrochemical reactions in multicomponent solid, liquid and gaseous materials under and challenging temperature and pressure conditions.

To address these challenges, we developed and demonstrated a compact, multifunctional optical fiber sensor assembled by femtosecond-laser-inscribed FBG and open-cavity FPI capable of being inserted into the central hole of commercial cells and of continuously sensing internal temperature and pressure during cell thermal runaway, after which the sensors can survive and be reused. The commercial battery selected for testing is an 18650 cylindrical lithium iron phosphate (LiFePO$_4$, LFP) cell with the rated capacity of 1530 mAh which is recognized as possessing superior safety performance (thanks to the strong P-O covalent bond) over other chemistry so that it is very successful in the energy storage market. Despite this, there remain safety issues with LFP cells, as evidenced by reported devastating accidents involving overheated LFP batteries[31,32]. The implanted fiber sensor offers a stable and reproducible correlation between the complex cell reactions and the optical signals. Further analysis shows that the sensor offers a scalable safety solution that can warn of impending thermal runaway well before catastrophic safety venting of the cell. This operando "lab-on-fiber" measurement tool provides crucial additional capabilities to battery monitoring methods, identifying the onset of unprovoked thermal runaway before it becomes dangerous and therefore enables a safety early warning to be issued and/or automatic shutdown.

## Results

### Principle of lab-on-fiber sensing technologies

Fig. 1a and b shows the configuration and operation principle of the optical fiber sensor implanted inside a commercial 18650 cell. The sensing probe is assembled from two sections, one is a FBG (0.8 mm in grating length) and the other is a FPI (0.1 mm in open cavity length). There is a 10 mm fiber gap between them to avoid optical interference. The resulting total length of the sensor is about 12 mm with a 125 μm uniform diameter. The sensing probe is spliced at the end of a standard communication single-mode optical fiber and located into the central hole of the cell. The fiber sensor is interrogated with a broadband light source as shown in Fig. 1c.

Then Fig. 1d–f present the SEM imagines of FBG in cross section view (left) and a side view (right), its principle and spectral response characteristics, respectively. In detail, the FBG reflects a narrow resonance (the Bragg resonance, $\lambda_B$) from broadband infrared light injection. $\lambda_B = 2n_{eff}\Lambda$, where $n_{eff}$ and $\Lambda$ denote the effective refractive index of the core-guided mode of the fiber and the period of the grating, respectively. Temperature variations change both $n_{eff}$ and $\Lambda$ by means of elasto-optic and thermo-optic effects and therefore shift the wavelength of the FBG resonance $\lambda_B$ [34–36]. Our experimental tests demonstrate that the wavelength shift of $\lambda_B$ shows linear response (R$^2$ = 99.9%) to temperature over the temperature range of 25 ~ 600 °C. On the other hand, FBGs in silica fibers are insensitive to pressure, especially for pressures below 2 MPa, as encountered in a working cell, see Fig. S1. Sensitivity of the FBG to strain, as caused by fiber bending, can be eliminated ensuring the fiber remains unbent by appropriately fixing the sensor at one end where it enters the cell and suspending the other end within the cell.

Fig. 1g–i show the SEM structure, sensing principle and spectral characteristics of the FPI sensor, respectively. In detail, the reflected light spectrum from the same input light presents periodic resonances due to interference between reflected light from the surfaces of M$_1$ and M$_2$. According to the phase matching condition of the FPI, when the initial phase of the interference $\varphi_0 = 0$, if we track the $k^{th}$ order spectral dip wavelength ($\lambda_{FP}$) of the FPI, its phase $4\pi n_m L/\lambda_{FP}$ keeps a constant value of $(2k+1)\pi$ [37], and the value of $\lambda_{FP} = 4n_m L/(2k+1)$, where $k$ is an integer, $L$ is the distance between the two reflection planes and $n_m$ is the refractive index of the medium between the planes[33]. Obviously, pressure may change the gas refractive index $\triangle n_m$ inside the open cavity (but not so much the cavity length at the pressures tested)[38,39]. Experimental tests demonstrate that the shift $\lambda_{FP}$ shows a highly linear response (R$^2$ = 99.9%) to pressure change range of 0 ~ 2 MPa. On the other hand, it was demonstrated that the FPI is insensitive to temperature change. For a large temperature change from 25 to 600 °C, the wavelength shift of the FPI is less than 0.3 nm, corresponding to a negligible temperature sensitivity of 0.5 pm °C$^{-1}$, see Fig. S2.

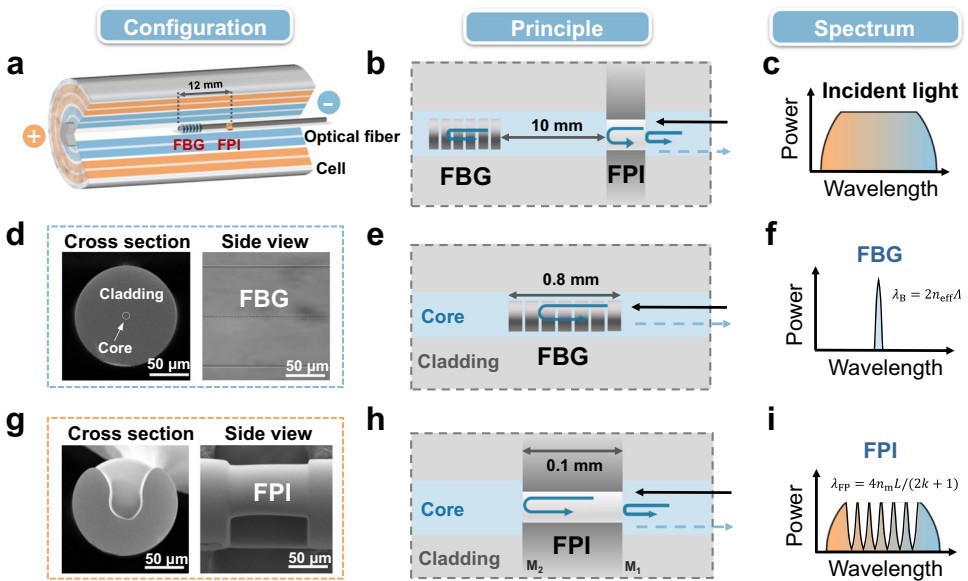

**Fig. 1 | Principle of combined FBG/FPI sensor for simultaneous temperature and pressure monitoring in the cell. a** The position of sensor in the battery. **b** The cascade structure of FBG-FPI sensor. **c** the incident broadband light spectrum. **d**–**f** The FBG configuration (left: cross section, right: side view), sensing principle (the periodic refractive index modulation of FBG reflects a narrow resonance) and its spectrum (narrow Bragg wavelength $\lambda_B$), respectively. **g**–**i** the FPI configuration[33] (left: cross section, right: side view), sensing principle (the open cavity interferes the light in between two surfaces $M_1$ and $M_2$) and its spectrum (several relatively broadband resonances), respectively.

## Characterization of optical fiber sensor and cell performance

Fig. 2 shows the $T$ and $P$ calibration results of the FBG and FPI components of the sensor. The FBG between 25 °C and 600 °C is displayed in Fig. 2a. The zoomed inset presents wavelength shift of the FBG resonance $\lambda_B$ versus temperature, and Fig. 2b shows the highly linear relationship between $\lambda_B$ and $T$ with a sensitivity of 10.3 pm °C⁻¹. The FBG temperature response has been carefully calibrated with a commercial thermocouple over large temperature range, see Fig. S3. Over the same temperature range the wavelengths of the FPI spectral fringes remain almost constant, as discussed above. Fig. 2c presents the FPI response to pressure over a range of 0 to 2 MPa. Precise qualification of pressure has been achieved by monitoring the wavelength shift of the FPI broad resonances (here we selected the FPI resonance at 1565 nm). Fig. 2d shows that the FPI offers a highly linear relationship to pressure with a sensitivity of 4188.4 pm MPa⁻¹, while the FBG is insensitive to pressure change (sensitivity of −5.6 pm MPa⁻¹). A further issue that should be addressed is that during cell thermal runaway, different kinds of gases will be generated. Fortunately, the FPI presents nearly identical pressure responses for different gases (see Fig. S4). The variation of FPI pressure sensitivities to the different gases is less than 0.5% ($P$ sensitivities change from 4.17×10³ to 4.19×10³ pm MPa⁻¹). This is an important finding for pressure measurement under such complex gaseous environments. Over the whole thermal runaway reaction, we found a negligible influence of electrolyte composition on the FPI's cavity resonances, thanks to the small cavity length of tens of micrometers and special tube-packaging method (see Fig. S5a).

After the above sensing calibration, the FBG-FPI sensor was implanted into commercial 18650 LFP cells (Fig. S5b and the Methods section provides the details implanting processes). Then, we evaluated whether the cell performance was influenced by optical fiber sensor implantation. Fig. 2e and f demonstrate that such influence is very limited, both for the different electrochemical charging conditions (from 0.5 C to 2 C) and for a cell lifetime test (100 cycles at 2 C charging rate, making the state of health of the cell drop down to 80%). For the specifications of 18650 LFP cells used here, please see Table S1.

As a final test of the proposed sensor, we showed that it offers excellent reproducibility before and after cell thermal runaway, as shown in Fig. S6. We attribute the outstanding performance of the sensor to the commercially available silica fibers (with Ge doping[40]) and the standard manufacturing technique for mass production (femtosecond-laser-inscribed FBG and thermal drawing open-cavity fiber, both offer excellent performances at high temperature over 1000 °C[40,41]). The detailed fabrication processes of the FBG and FPI are shown in Figs. S7, S8, S9 and S10.

## Operando monitoring of the cell during thermal runaway

The experimental setups and their logical relationship for cell thermal runaway characterization are depicted in Fig. 3a and b. The key parameters to be monitored include the internal temperature and pressure (measured by our FBG & FPI optical fiber sensor), surface temperature (by the thermocouple), voltage (by the battery testing system), and mass loss (by the mass balance). The thermal runaway is triggered by a cylindrical heater with the same size of the cell that tightly attached to the cell (Fig. 3c). Such heating is stopped after the cell thermal runaway has been triggered (Fig. 3d). The heater mimics a cell experiencing thermal runaway in a cell pack, thus affecting the adjacent cell, an approach widely used in cell safety assessment. Fig. S11 provides all the heating conditions for 100%, 50% and 0% SOC (state of charge) cells.

As shown in Fig. 4, three typical SOCs (100%, 50% and 0% SOC) were selected for cell thermal runaway analysis. Fig. 4a, c and e present the real-time responses of the cell's internal temperature and pressure, mass loss and output voltage during thermal runaway. Side by side, Fig. 4b, d and f show corresponding images recorded at each critical moment, classified according to four stages: safety venting (I), incubation period (II), thermal runaway (III), and cooling down (IV). The meaning of codes 1 to 8 and their logical relationships are explained in Fig. 4g. Detailed measured $T$ and $P$ data are listed in Table S2. A video for the thermal runaway processes of the cell with 50% SOC were recorded, as shown in the Supplementary movie 1.

**100% SOC cell thermal runaway.** *Stage I (safety venting)* after the heater is turned on, the surface temperature of the cell quickly increases by heat conduction from the heater. The surface

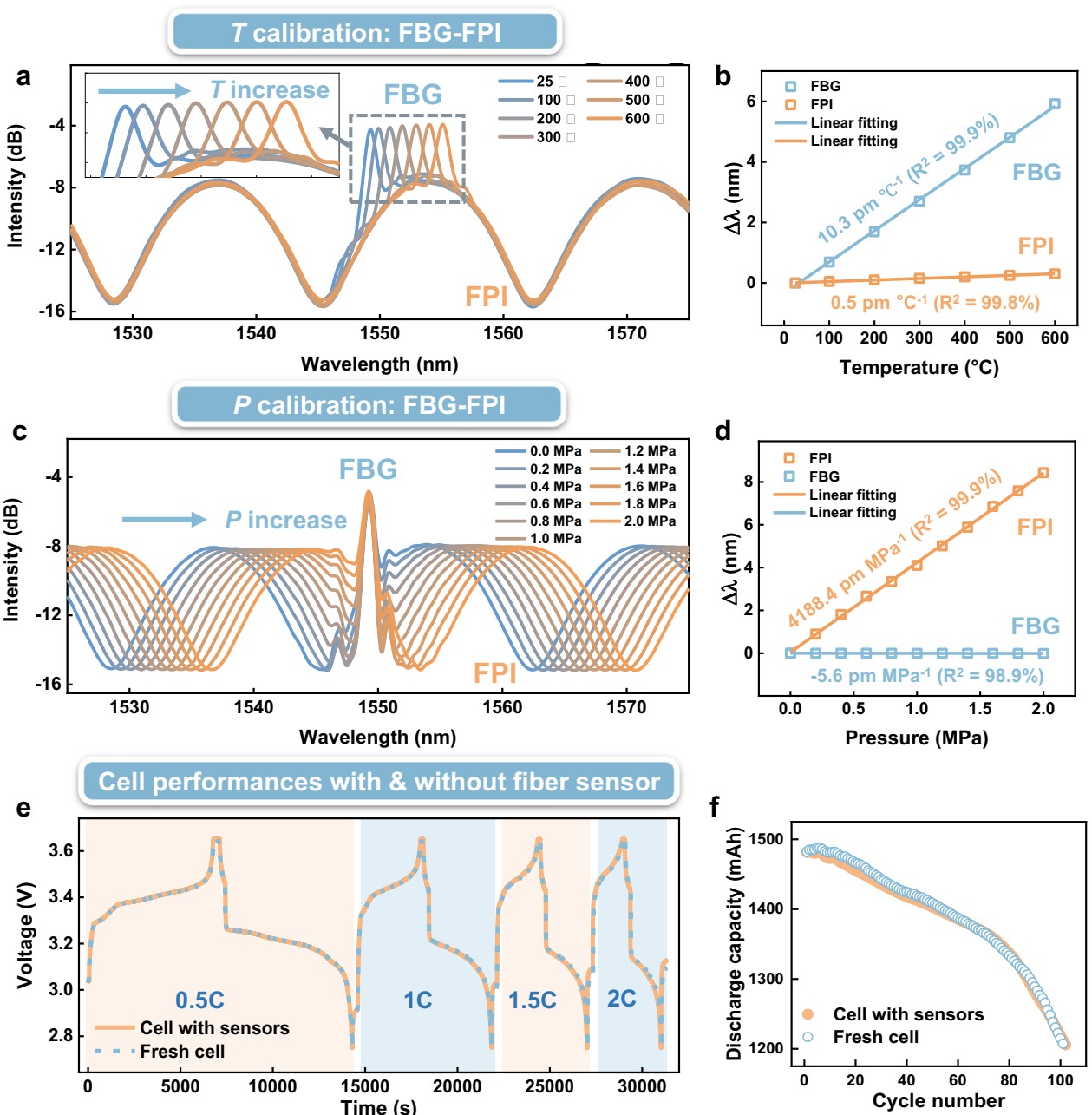

**Fig. 2 | Characterization of FBG-FPI sensor and evaluation of cell's performance with optical fiber implanted. a, b** The FBG-FPI spectral and wavelength responses to temperature over the range from 25 to 600 °C, in which the FBG displays linear temperature sensitivity (see the zoomed FBG spectra inset), while the FPI shows insensitivity to temperature. **c, d** The FBG-FPI spectral and wavelength responses to pressure over the range of 0 - 2 MPa, in which the FPI shows a linear pressure sensitivity (in air), while the FBG shows insensitivity to pressure. **e, f** Demonstrate that the performance of cell is unaffected after implanted with optical fiber sensors. The cells used here are commercial 18650 lithium iron phosphate cells, the charging conditions are varied from 0.5 C to 2 C and the long-term test lasts for 100 cycles at 2 C ("C" indicates "C rate", which is the ratio of charge/discharge current to rated capacity).

temperature is generally higher than the internal temperature due to the lower thermal conductivity[42]. While the internal pressure remains nearly constant at 0.1 MPa for over 100 s (a well-balanced pressure), as the temperature further increases (reaching approximately 70 °C), the internal pressure also starts to increase as a consequence of electrolyte evaporation[43]. Then, the SEI layer begins to decompose, and $O_2$, $CO_2$ and $C_2H_4$ are successively released[44], leading to an obvious pressure increase. Subsequently, an internal short circuit is initiated, as evidenced by the sharp voltage drop to 0 V[45]. It is interesting to find that there is an obvious voltage drop accompanied by a 20 °C temporary jump lasting for approximately 40 s (the zoomed inset of Fig. 4a) and

then recovering back to the initial increase rate. We interpret this phenomenon to be Joule heating, which is associated with current flowing through the internal short circuit. Such a presumption is supported by the fact that the change in cell surface temperature (6 °C) is much smaller than the internal temperature jump (20 °C). Despite the internal short circuit, thermal runaway still does not occur until approximately 53 s later, indicating that major parts of the energy are still stored inside the cell[46]. Safety venting occurs as more and more gases are generated inside the cell, causing a rapid increase of the internal pressure to 1.79 MPa until the pressure exceeds the threshold of the safety valve. ***Stage II (incubation period)*** is identified by the

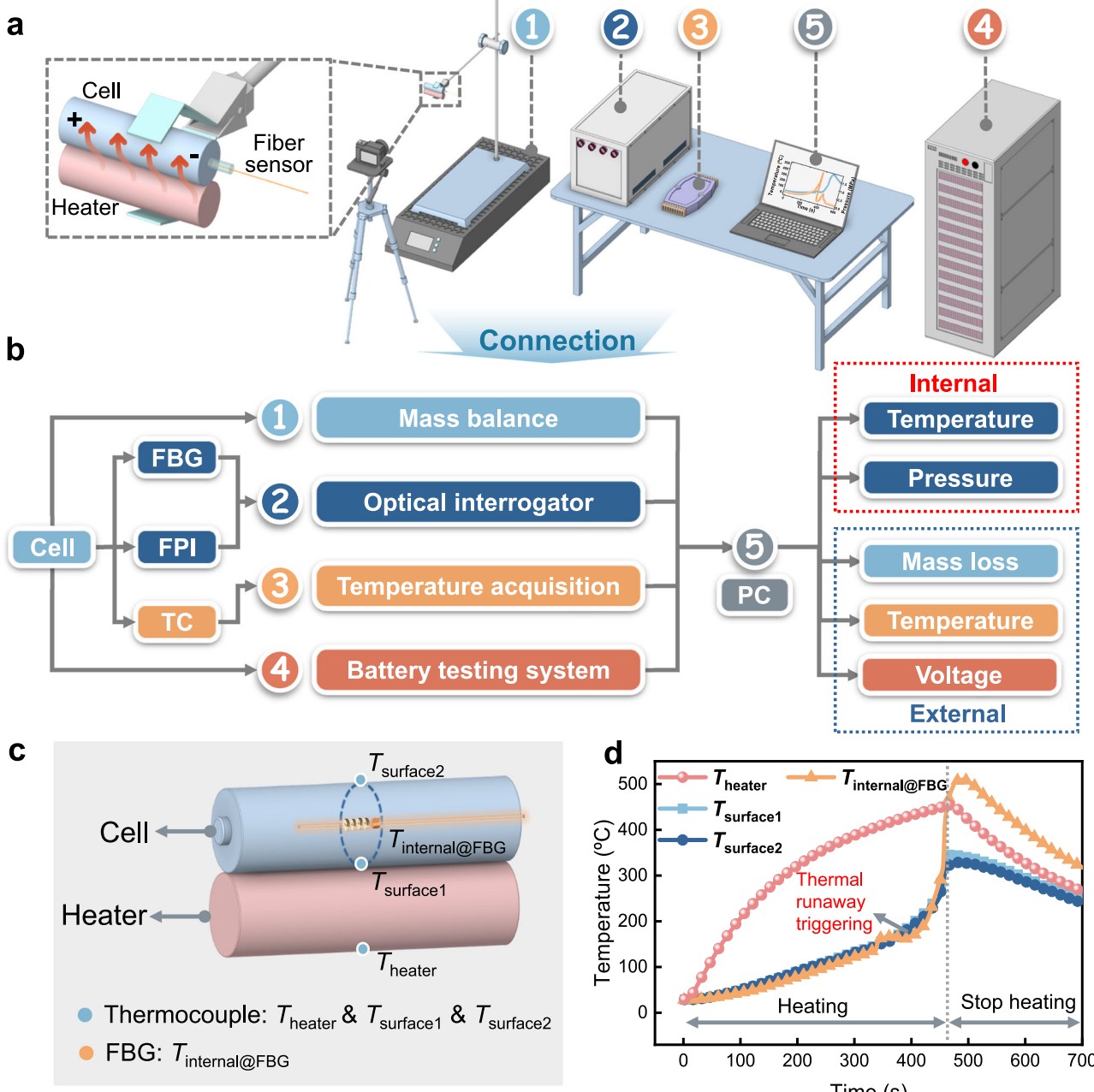

**Fig. 3 | Experimental setups for cell thermal runaway measurement.**
**a** Experimental setups ① mass balance for cell mass loss monitoring, together with a heater and cell fixing setup, ② multichannel optical spectrum interrogator for FBG-FPI optical signal monitoring, ③ electronic data reader for the thermocouple, ④ battery testing system wired connected with cell for current and voltage readout, ⑤ laptop for all data acquisition and analysis (internal temperature and pressures, mass loss, surface temperature, voltage). **b** Logical relationship and functions for the above experimental setups. **c** Detailed positions of surface thermocouples ($T_{heater}$, $T_{surface1}$, $T_{surface2}$) and internal optical fiber sensor ($T_{internal@FBG}$). **d** Heating conditions used to trigger the thermal runaway (red curve shows a gradually increasing heater temperature, followed by a gradual decrease after the heater current is abruptly stopped once thermal runaway is initiated).

onset of mass loss and the sudden pressure drop to 0.1 MPa owing to instantaneous electrolyte vapor and gas release[47] (photographs in Fig. 4b). The cell then enters the incubation period. Close examination of the insert enlarged plot in Fig. 4a shows that the temporary small jump in temperature eventually diminishes during venting because of heat dissipation by the ejected materials from the Joule-Thomson effect[48] observed previously in classic thermal runaway tests[31,32]. Now, the cell is ready for thermal runaway. Exothermic chain reactions involving "anode-electrolyte-cathode" materials then begin to accumulate heat and generate gases again, as evidenced by an increase in the internal temperature increasing to a value significantly higher than

surface temperature because the generation of heat is dominated by internal electrochemical reactions and no longer driven by the external heater. A secondary and smaller pressure peak appears, as well as mass loss. **Stage III (thermal runaway)** is accompanied by violent white smoke ejection and continued internal temperature increase toward the maximum approaching 510 °C. The maximum internal-to-surface temperature difference ($\Delta T_{max}$) increases to over 180 °C, because the transient generation of heat that cannot rapidly dissipate. Then, the final Stage IV (cooling down) occurs when all possible reactions have completed, with no more mass loss and gas generation, and the cell temperature gradually recovers to ambient conditions.

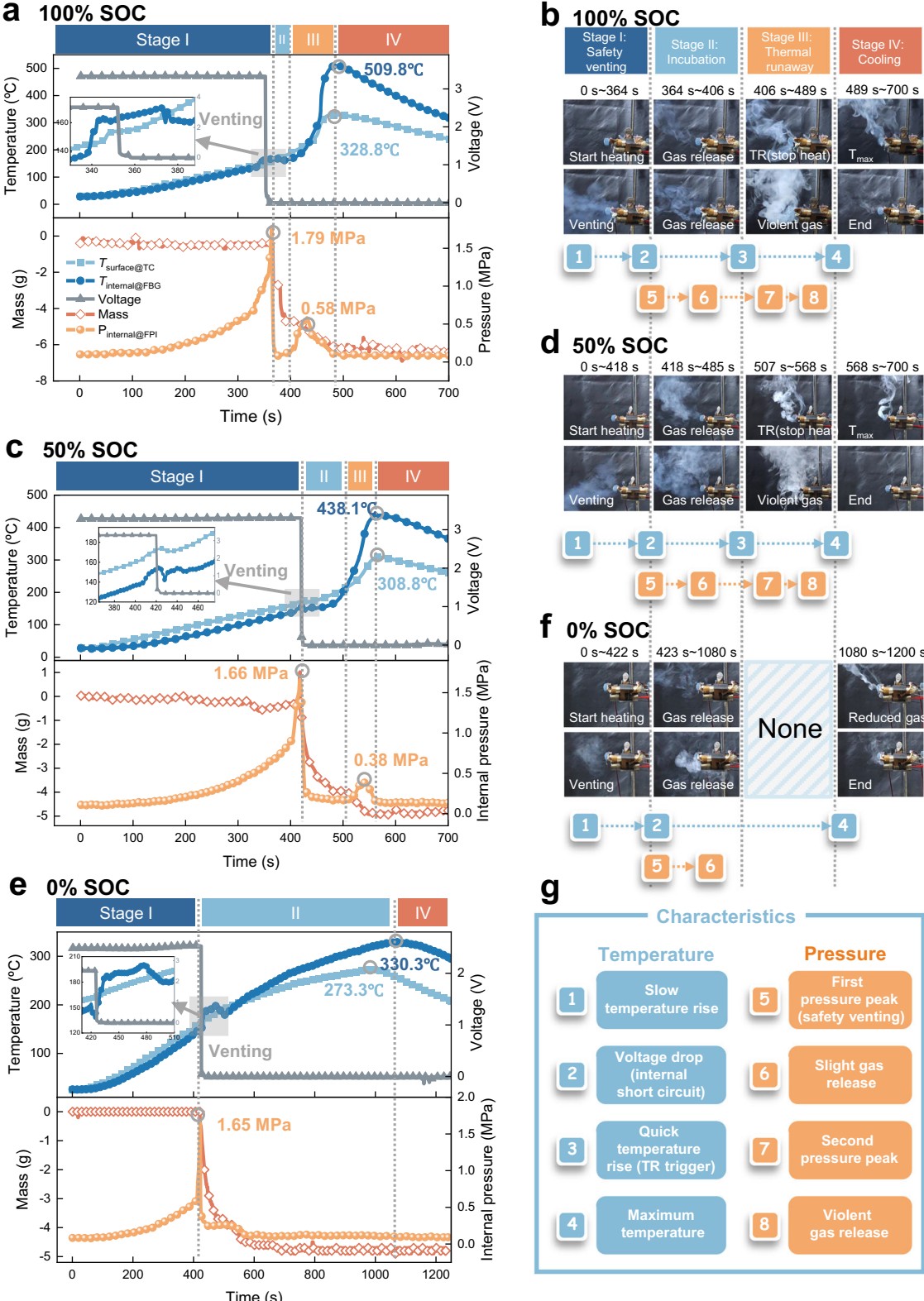

**Fig. 4 | Operando monitoring internal temperature and pressure during 18650 cells thermal runaway.** Thermal runaway characteristics of 18650 cells with 100% SOC (**a**, **b**), 50% SOC (**c**, **d**) and 0% SOC (**e**, **f**). **a**, **c**, **e** In situ internal temperature (dark blue line) and internal pressure (orange) of the cells during thermal runaway captured by implanted optical fiber sensor (gray circles highlight their peaks) and surface temperature read from thermocouples attached to the cell (light blue line). The voltage output (dark gray line) sharply drops at the occurrence of an internal short circuit, causing about 20 °C of temperature rise lasting for tens of seconds (light gray squares and their zoomed insets for all the SOC cases). The mass (red line) also suddenly drops with safety venting. **b**, **d**, **f** Photographs at moments of the cells critical reactions during thermal runaway, including safety venting (I), incubation period (II), thermal runaway (III), and cooling down (IV). The cell with 0% SOC does not contain Stage III (blue shading) because of insufficient energy to initiate thermal runaway. **g** characterization of all correlated steps, in which the blue-labelled ones relate to temperature and the orange ones corresponding to pressure ("TR" indicates thermal runaway).

**50% SOC cell thermal runaway.** The runaway develops and proceeds similarly to the 100% SOC case. However, the triggering of thermal runaway is delayed by 100 s and lower maximum of internal pressure (approaching 1.66 MPa) and temperature (approaching 440 °C) are observed as well as smaller mass loss. This is reasonable because the lower SOC implies that there is less lithium intercalation into the graphite anode and consequently less electrochemical energy storage than in the 100% SOC case.

**0% SOC cell without thermal runaway.** Unlike the thermal runaway stages for the 100% and 50% SOC cases, thermal runaway (Stage III) does not occur in the cell and the maximum internal temperature reaches only to 330 °C. Interestingly, there is no (at least not clear) second pressure peak, due to lack of stored internal energy.

### Identification of reversible and irreversible conversion

In the absence of our proposed sensor, thermal runaway warning signals are highly dependent on observation of the gas release due to safety venting[49] and voltage drop induced by the internal short circuit[50]. These warning signals currently occur too late, as by the time they are observed, the cell has already been irreversibly damaged. Great benefit could be gained if it were possible to generate an alarm signal at the very beginning of an adverse cell thermal event, before the cell undergoes irreversible damage.

We approached this challenge by analyzing the time derivatives of the cell's internal $T$ and $P$, as illustrated in Fig. 5a and b. The insets of Fig. 5b (two vertically expanded versions of the graph segments in the red dashed frames) present the detailed response of differentiation of $T$ and $P$ with respect to time. For both of them, two different tendencies can be clearly found for time intervals labelled ① and ②, in which ① indicates a rapid temperature rise and a constant pressure at early times, whilst ② indicates an accelerated pressure rise with a stable temperature rise at later times. Fig. 5c, d and e distinctly present the derivatives of $T$ and $P$ during periods ① and ②, prior to safety venting versus time. For all tested SOCs (100%, 50%, 0%), the combined $T$ and $P$ derivative curves delineate "rhombus shaped" responses. Fig. 5f reveals the physical and chemical mechanism behind the stage ① and ② reactions. For state ①, the temperature rise comes from the heat conduction of the heater, which induces electrolyte evaporation and boosts the internal pressure. Then, a continuing temperature rise facilitates more electrolyte vapor and the onset of irreversible SEI decomposition (see Fig. S12b). As a result, a large amount of gas generation quickly increases the internal pressure at the expense of heat consumption and transfer to gases, as shown in stage ②. In between ① and ②, the "turning section" characterizes the enhanced electrolyte evaporation and the early onset of irreversible SEI decomposition at temperature around 70 ~ 80 °C (detailed data in Table S2) for all cases of SOCs. A safety warning range is thereby set to begin from electrolyte evaporation and to end with SEI decomposition, a range where irreversible reactions have not yet occurred and which guarantees the normal use of the cell before damage. As presented by Fig. 5c, d and e, these phenomena do not depend on the SOCs of the cells, further confirming that the "warning range" can be regarded as a general cell safety alarm indicator.

Fig. 6 presents an overview of the correlation between in situ $T$ and $P$ in Li-ion cell and its complex thermal runaway reactions, including electrolyte evaporation, SEI decomposition, separator melting, internal short circuit, safety venting, followed by drastic chain exothermic reactions (electrode & electrolyte interactions). A scalable solution for identification of reversible and irreversible conversion of cell's internal reactions has been demonstrated, by determining the turning section of the differential curves of temperature and pressure (see the inset of Fig. 6). Such method provides an early safety warning range before safety venting, facilitating the cell safety assessment and warning of thermal runaway. Finally, the classic characterization

results based on SEM-EDS and XRD methods, before and after cell thermal runaway, have been demonstrated and shown in Figs. S13–S15.

### Discussion

Using our proposed optical fiber sensor, we successfully achieve in situ, operando continuous monitoring and precisely decoding of the internal temperature and pressure of commercial Li-ion cell prior to and during thermal runaway, without disturbing the cell's operation. Furthermore, a stable and reproducible correlation between the cell thermal runaway and the optical response has been observed and precisely quantified. The sensor offers a scalable solution for the identification of a safety early warning range for thermal runaway from slope changes of the differential curves of temperature and pressure before safety venting. The proposed FBG-FPI optical fiber sensor offers excellent sensing characteristics and high reproducibility before, during and after cell thermal runaway. They are small size, flexible in shape, and offer electrical interference immunity and remote operation, and are amenable to standard manufacturing techniques for mass production.

With the development of clean-energy systems, ranging from electric cars to power plants, it is meaningful to realize a portable and cost-effective interrogator for in field measurement. By using a tunable laser as a source (for example, a compact VCSEL[51]) together with a photodiode as detector and an analog-to-digital converter (A/D) to obtain the desired data, traditional broadband light sources and expensive optical spectrum analyzers can be replaced. This can be achieved by setting the tunable laser's wavelength matching to the most sensitive spectral resonance of the sensor, and the wavelength shift of the resonance be monitored can be transformed to the optical power changes correspondingly (see Fig. S16).

Through the true multidisciplinary efforts between of converging electrochemistry, sensing technologies and data science, new types of embedded lab-on-fiber sensors will allow the continuous monitoring of battery health and safety status, and will promote more reliable battery systems[52]. Various key parameters including temperature, pressure, refractive index, gases and ions concentration can be monitored simultaneously in an operando way over one optical fiber, at multi-positions of the battery. This provides theretofore unrealizable crucial capabilities in safety of operation as well as complementary information regarding battery state of health and evolution. Given the potential for optoelectronic integration of the components needed, it is not unthinkable to envision widespread use of these techniques in mass market applications, such as electric vehicles.

### Methods

#### Fabrication of the FPI

A section of open-cavity fiber is spliced to a standard single-mode fiber (SMF), which is cut using a common fiber cutter (FL-21F, JILONG) with the help of a microscope (AXIO SCOPE A1, ZEISS). The cut open-cavity fiber is then spliced to another SMF using manual alignment, and the other end is pinched off directly to avoid unwanted reflections. The fabrication of the FPI is shown in Fig. S7. It should be mentioned that the cavity length of each FPI varies slightly due to the manual steps involved in their fabrication, but these small variations in cavity length have no effect on the pressure sensitivity, as shown in Fig. S8.

#### Fabrication of the FBG

The FBG was fabricated 800 nm light from a Ti: sapphire femto-second laser system (Coherent, using Libra-USP-HE), as described in Fig. S9. The laser beam is focused by a ZEISS, 40X, 0.75NA microscope objective, producing a spot diameter smaller than 1 μm. The optical fibers are fixed on a high-precision motorized displacement stage (Newport, XMS 100-S, precision <40 nm, stroke 100 mm), which enables the fibers to be moved with high precision over short distances. During the fabrication of the FBG,

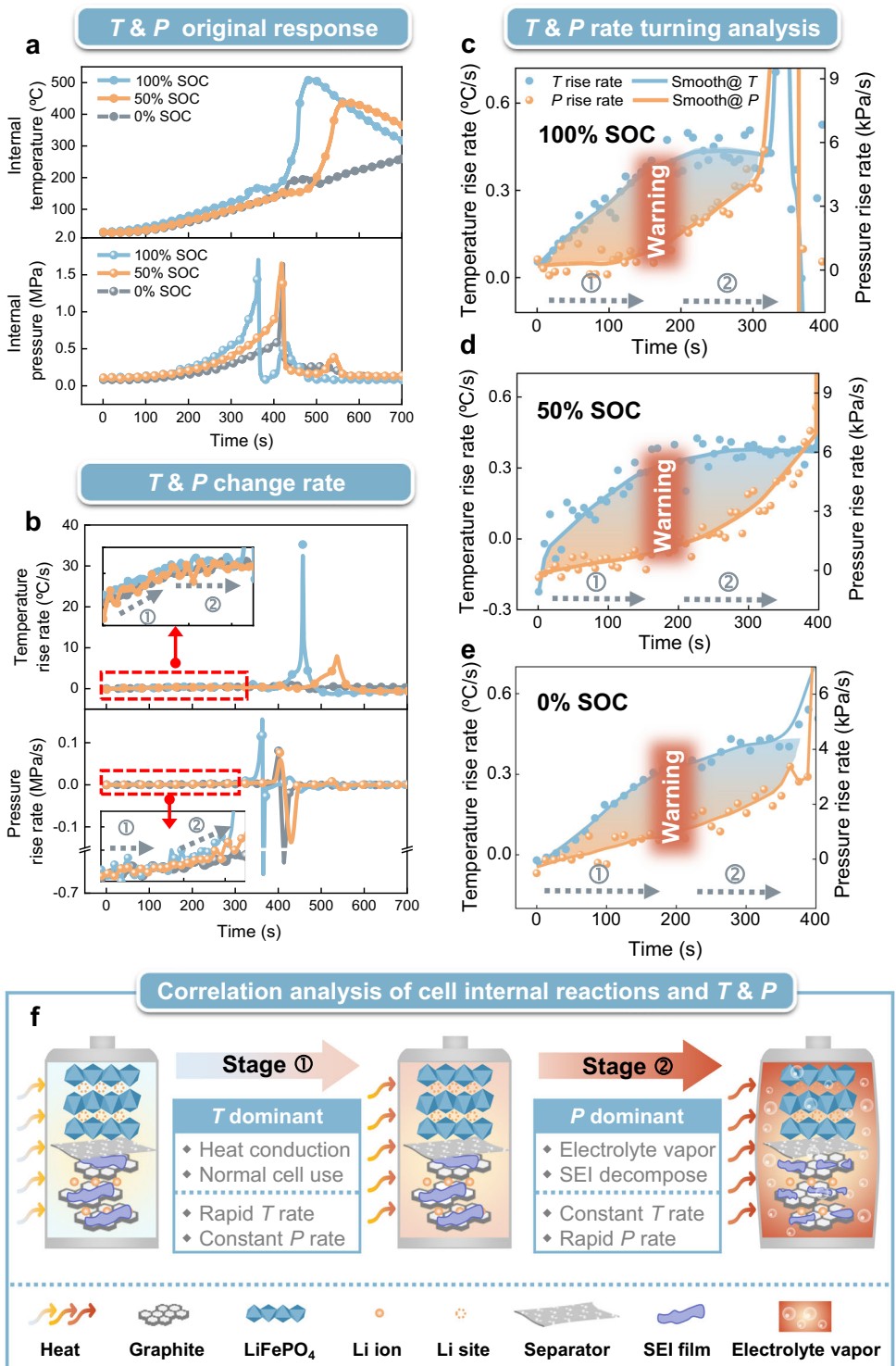

**Fig. 5 | Establishing a safety warning range by detection of the switch from reversible and irreversible reaction through temperature and pressure time derivatives. a** Internal temperature and pressure responses of the cells with 100%, 50% and 0% SOC during thermal runaway. **b** Rates of increase of internal temperature and pressure with time. Insets magnify the vertical scale during the two response periods ① and ② prior to the internal short circuit. **c**, **d** and **e** The combination of temperature and pressure rise rates forming a rhombus-shaped characteristic response in each of the 100%, 50% and 0% SOC cell cases. ① shows a rapid

temperature rate increase while the pressure rate remains constant, and ② shows the opposite tendency. Between ① and ② the turning section identifies the parts of the derivative curves used to generate onset of a safety warning range (red squares), regardless of the SOCs. **f** Unveils the inherent physical-chemical mechanism of reversible and irreversible reactions between stages ① and ②. A warning can be issued as the cell suffers from healthy reversible variations (electrolyte evaporation) but before harmful irreversible reactions occur (SEI decomposition).

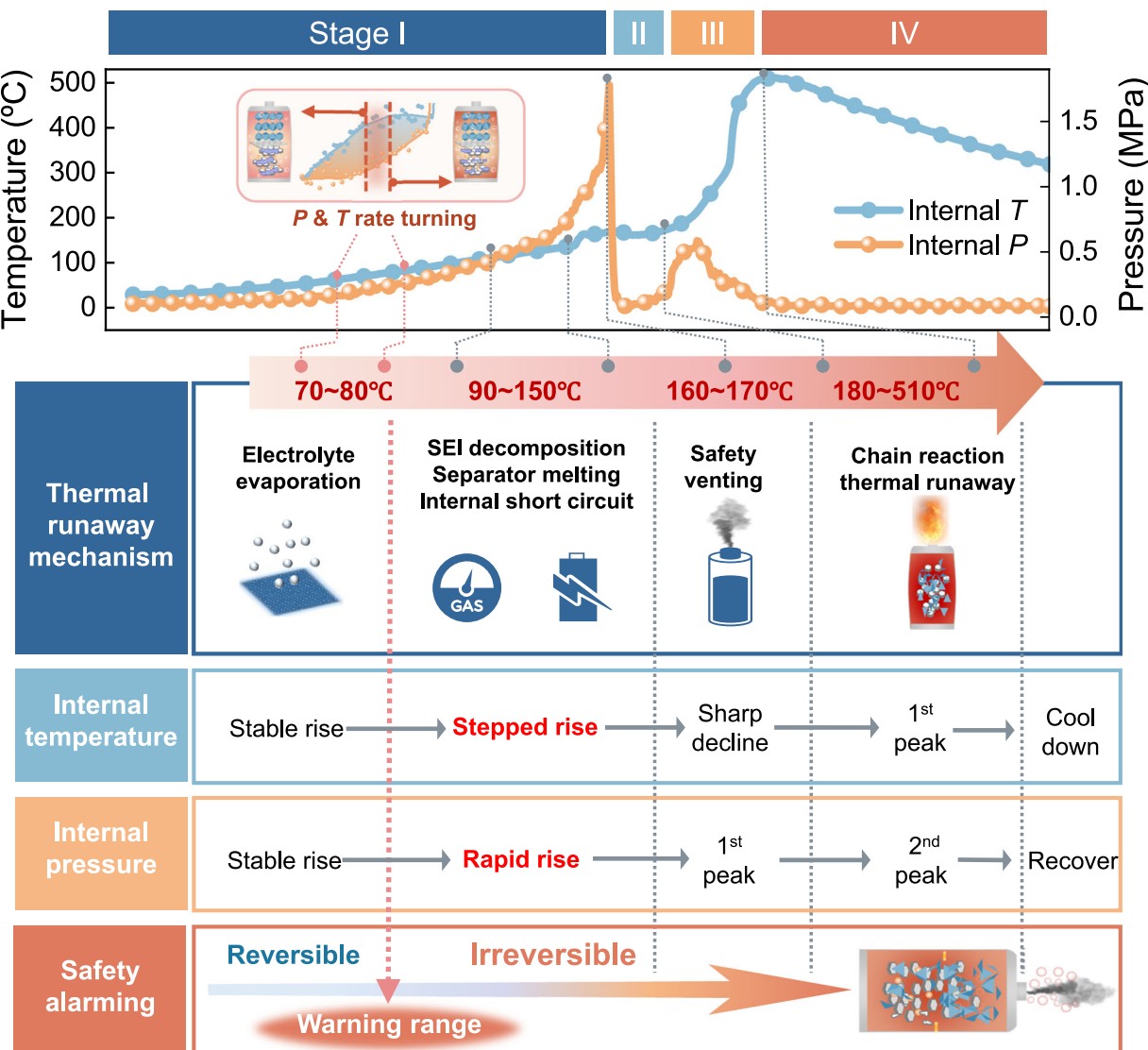

**Fig. 6 | Mechanism of thermal runaway and establishing early warning of impending cell damage.** Operando temperature and pressure monitoring throughout cell thermal runaway, followed by decoding the complex thermal runaway reactions and figuring out their correlations from measured internal temperature and pressure. The safety warning before cell venting can be figured out by identifying the turning section of "rhombus-shaped" derivative curves of $T$ and $P$ (above inset red block diagram), regardless of the cells' states of charge.

the reflectance and transmission spectral properties of the FBG are monitored in real time using an optical spectrum analyzer (YOKOGAWA, AQ6380). A broadband light source (SC-5-FC/APC, Wuhan Yangtze Soton Laser Co., Ltd) is used to provide a target wavelength window of emitted power.

**Fabrication of the FBG-FPI**
We fabricate the open cavity FPI at first and then inscribe the FBG on the downstream side using a femtosecond laser with a separation of 10 mm between the two. The details of the fabrication processes can be found in Fig. S10.

**Calibration of FBG-FPI sensors**
The temperature calibration of the FBG-FPI is conducted in a high-temperature tube furnace from 25 °C to 600 °C, with a step of 100 °C and a rest time of 2 h at each step. The pressure calibration of the FBG-FPI sensor is carried out in a cylindrical pressure chamber. The pressure is recorded from 0 MPa to 2 MPa in 0.2 MPa increments using a pressure gauge (CONST 211). A maximum pressure of 2 MPa since it is recognized pressure venting threshold for this commercial 18650 LFP cell. Two

kinds of spectral interrogators are used for different experimental environments. An optical spectrum analyzer (AQ6380, YOKOGAWA, wavelength accuracy of 1 pm, spectral range of 1500–1600 nm) is used in a clean lab for optical fiber sensor fabrication, calibration and packaging. Another optical spectral interrogator (FS22SI, HBM, wavelength accuracy of 1 pm, spectral range of 1500–1600 nm) is used for battery thermal runaway measurements, as provided in Table S3.

**Principle of decoding temperature from FBG**
Supposing an isothermal condition, the Bragg wavelength variation ($\Delta\lambda_B$) upon strain and temperature changes can be expressed as Eq. (1)[34],

$$\Delta\lambda_B = \lambda_B \left[ (1 - P_\varepsilon)\varepsilon + (\alpha + \xi)\Delta T \right] \tag{1}$$

where $P_\varepsilon$ represents the effective photoelastic constant, $\varepsilon$ and $\alpha$ are the strain and thermal expansion coefficients, respectively, of the optical fiber, $\xi$ denotes the thermo-optic coefficient and $\Delta T$ indicates the temperature variation. The FBG acts as a reflector to decode temperature and strain variations in the environment surrounding the FBG by monitoring the wavelength shift of $\Delta\lambda_B$. In this experiment,

the FBGs are suspended in a hole drilled in the cell and parallel to its axis. Care is taken to avoid any contact between the sensor and the cell, so that the effect of strain variation on the Bragg wavelength can thereby be neglected. Hence, the shift of the central wavelength is merely caused by temperature change.

## Principle of decoding pressure from FPI

When light is incident at a certain angle through two parallel mirrors, it is reflected and refracted several times between the two surfaces, so that any two beams with the same optical path difference, the same frequency and a constant phase difference will interfere and form a corresponding interference spectrum. According to Fresnel reflection theory, the two mirrors ($M_1$, $M_2$) formed at the connection interface between the open cavity fiber and the SMF have a very low reflectivity ($R \approx 3.5\%$ for an air cavity). Therefore, the FPI formed with an air cavity can ignore the multiple reflections between the two mirrors, and the FPI based on the open cavity fiber can be considered a two-beam interferometer. Its FPI spectrum can be expressed as[33]:

$$I(\lambda) = I_1 + I_2 + 2\sqrt{I_1 I_2}\cos\left(\frac{4\pi n_m L}{\lambda} + \varphi_0\right) \quad (2)$$

where $I(\lambda)$ is the optical intensity of the FPI spectrum, $I_1$ and $I_2$ are the reflection intensities corresponding to the two mirrors ($M_1$, $M_2$) from the FPI, $\varphi_0$ is the initial phase difference of the two beams, $L$ is the cavity length, $n_m$ is the refractive index of the medium in the open cavity, and $\lambda$ is the free space wavelength.

The wavelength spacing between two adjacent dips of the reflected spectrum is known as the free spectral range ($FSR$),

$$FSR_{FP} = \frac{\lambda_1 \lambda_2}{2 n_m L} \quad (3)$$

where $\lambda_1$ and $\lambda_2$ are wavelengths of two adjacent dips in the FPI reflection spectrum.

For the FPI pressure sensor with the open cavity structure, the gas refractive index change caused by the pressure change is calculated according to Eq. (4)[38,39],

$$n_m = 1 + \frac{2.8793 \times 10^{-9} \times P}{1 + 0.003661 \times T} \quad (4)$$

where $P$ and $T$ represent pressure (Pa) and temperature (°C), respectively.

When FPI is used for pressure measurements, the dip wavelength $\lambda_{FP}$ of the interferometric spectrum varies with pressure. From this, we obtain the gas pressure sensitivity as

$$\frac{d\lambda_{FP}}{dP} = \lambda_{FP}\left(\frac{1}{L}\frac{dL}{dP} + \frac{1}{n_m}\frac{dn_m}{dP}\right) \quad (5)$$

where $\frac{1}{L}\frac{dL}{dP}$ represents the change in cavity length with pressure and $\frac{1}{n_m}\frac{dn_m}{dP}$ represents the change in refractive index of the gas with pressure.

The cavity length $L$ of the PFI can be regarded as constant when the pressure changes. Therefore, Eq. (5) can be simplified as follows:

$$\frac{d\lambda_{FP}}{dP} = \frac{\lambda_{FP}}{n_m}\frac{dn_m}{dP} \quad (6)$$

where $\frac{dn_m}{dP}$ is the rate of change of the refractive index of the gas with pressure.

Eq. (6) shows that the wavelength shift of the interference spectrum is linearly related to the change in the refractive index of the gas in the cavity caused by the change in pressure. Thus, the pressure change inside the cell is monitored by the shift of the FPI peak wavelength.

## Cell preparation

The fresh 18650 LFP cells are first charged at constant current-constant voltage (CC-CV) and discharged at constant current protocols with voltage ranging from 2.5 V to 3.65 V three times for SEI formation, with the current dropping to 0.01 C during the CV charge stage. Three minutes of relaxation are conducted between every cycle as well. The cells are prepared for drilling in discharged state (0% SOC) to reduce the risk during drilling. The modified cells are firstly charged to 100% SOC and then discharged to the prescribed SOCs (50%, 0% SOC) to ensure the performance of each modified cell. The cell specifications from the manufacturer are listed in Table S1.

## Implanting optical fiber sensors into 18650 cells

We drilled a 1 mm hole at the central position of the cell negative electrode in a glove box, ensuring no exposure to air, no damage and no internal short circuit are created in the cell. Then, we package the optical fiber sensor into a ceramic tube (made of $Al_2O_3$). Finally, we inserted the packaged sensor into the battery through the hole, positioned the sensing probe in the middle section of the cell and sealed the hole with epoxy resin which was cured for 24 h. Photographs of all steps of the implantation process are provided in Fig. S5b.

## Electrochemical tests

The 18650 LFP cells with/without optical fiber are first conducted on successive charge–discharge cycles at four charge–discharge rates of 0.5 C, 1 C, 1.5 C and 2 C to validate the rate performance after implanting sensors. Subsequently, cells are applied to continuous charge–discharge protocols at 2 C over several cycles until the state of health falls below 80% to further verify the cycle performance after implanting FBG and FPI. The charge rate of 2 C is chosen because once the unaffected cell performance is guaranteed at high rates, it will be fine at lower rates. All of these electrochemical tests were carried out by a Neware battery test system (CT-4008T-5V12A-S1, Shenzhen, China) within a temperature-controlled oven.

## Thermal runaway tests

The thermal runaway tests are implemented in an explosion-proof chamber with an exhaust hood to collect the smoke and gas. The 18650 LFP cell with implanted FBG and FPI is connected to the battery testing system to record voltage during thermal runaway, with a thermocouple attached on its surface. The battery is bound with iron wire to a cylindrical heater of the same dimensions as the battery, and the whole is clamped onto an iron frame. The cell is connected to the battery testing system to record voltage during thermal runaway. The FBG and FPI are connected to the optical spectrum analyzer, the thermocouples are attached to the data acquisition equipment (ICPCON I-7018), the heater is linked to a power conditioner to maintain the heating power at 100 W, and heating is stopped once thermal runaway is triggered (for 100% and 50% SOC), as evidenced by violent gas release. For 0% SOC, heating is stopped at the end of the test with no thermal runaway observed. All the optical and electronic data were collected by a laptop computer, and all the thermal runaway processes were recorded by a high-definition camera outside of the chamber. The physical map during and after thermal runaway is provided in Fig. S5c.

## Error assessment of the FBG temperature measurement

A commonly used thermocouple and a FBG were implanted into the 18650 cells simultaneously to validate the internal temperature response provided by the FBG during normal cycling and thermal runaway as shown in Fig. S3. The Pearson correlation coefficient between the thermocouple and FBG is over 99.8% both during normal

cycling and thermal runaway, displaying the highly consistent temperature response of the FBG.

## Postmortem methods

The postmortem technique involves disassembling the cell to collect electrodes for in-depth characterization (SEM–EDS, XRD and DSC, as shown specifically in Figs. S12–S15). The fully charged cells before and after thermal runaway are first dismantled in a fume hood to obtain the internal jellyroll. Afterwards, the jellyroll before thermal runaway is shifted to an argon-filled glove box, unrolling to separate the positive and negative electrodes. Several pieces of positive and negative electrodes were cut by ceramic scissors and washed three times with dimethyl carbonate (DMC). These pieces were dried in a glove box after 24 h of soaking with DMC in preparation for characterization. In contrast, the electrode materials have mostly lost contact with the current collector to become fragments on the one hand and have been unaffected by oxygen on the other hand after thermal runaway. These cells are directly unrolled in a fume hood to obtain positive and negative materials, and the physical map is provided in Fig. S5c. The subsequent steps are identical to those of cells before thermal runaway.

## Data availability

The authors declare that the main data supporting the findings of this study are available within the article and its Supplementary Information files. All other relevant data supporting the findings of this study are available from the corresponding author on request.

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

## Acknowledgements

This work is supported by the National Natural Science Foundation of China (No. 62035006, No. U2033204), the Youth Innovation Promotion Association CAS (No. Y201768), the Guangdong Outstanding Scientific Innovation Foundation (No. 2019TX05×383), the National Postdoctoral Program for Innovative Talents of China (No. BX20220286), the National Postdoctoral Science Foundation of China (No.2022M723040), the Program of Marine Economy Development of Natural Resources of Guangdong Province (No. GDNRC [2023]23). We would like to thank Prof. XG Qiao, Prof. RH Wang, Dr. FY Chen for their supports on optical fiber devices, and Dr. GY Gao, Dr. HM Zhou and Mrs. TT Fang for their helps in the XRD, SEM and DSC experiments.

## Author contributions

T.G. and Q.S.W. conceived the idea, supervised the project and analyzed the data. W.X.M., C.D.W. and P.J.L. carried out the thermal runaway experiments, W.X.M. conducted the thermal runaway data analysis and cell characterization. Z.L., Y.B.L., X.D.X., X.B.X. and X.L.H. fabricated sensors and carried out optical experiments. Z.L. and Y.B.L. provided thermal runaway optical data decoding. Theoretical analyses were done by C.W., J.H.S., H.Y.T., G.Z.X. and J.A. All authors contributed to the preparation of the manuscript. W.X.M. and Z.L. equally contributed to this work.

## Competing interests

The authors declare no competing interests.
