## [Peer review file · Nature Communications]

REVIEWER COMMENTS

Reviewer #1 (Remarks to the Author):

Review: Operando monitoring of thermal runaway in commercial lithium-ion cells via advanced lab-on-fiber technologies.

The manuscript discusses the use of compact fiber optic sensors inside of commercial 18650 Li-ion batteries for monitoring internal temperatures and pressures during thermal runaway. This allows for more direct and accurate monitoring of cell characteristics during testing, and can provide early indication of thermal runaway in cells during operation. The authors give a mostly thorough explanation of the sensor insertion and subsequent testing, as well as analysis of results.

The authors employed the fiber optics sensors Fabry-perot interferometer (FPI) in combination with the fiber bragg gating (FBG) for operando monitoring of the cell thermal runaway reactions in cylindrical lithium ion phosphate batteries. The single sensor is used to monitor the temperature and pressure during the thermal runaway using the Bragg wavelength variation. The pressure change is then decoupled using the peak wavelength shift of the interferometric spectrum of the FPI which is regarded as a function of pressure only. Detecting the turning sections of the time derivative curves of the pressure and temperatures are used in establishing a safety warning signal. The study would help in the battery health monitoring and generating an alarming signal much before the irreversible damage takes place.

However, the manuscript would be improved by addressing the following comments.

Comments/questions:

1. What is the manufacturer-rated lifetime of the LFP cells used in the testing? Normally LFP cells are expected to last hundreds or sometimes thousands of cycles, but the capacity fade in Figure 2 (f) shows the cells lose ~20% of their capacity in only 100 cycles. It would be helpful to have a table that lists the cell specifications provided by the manufacturer.
2. Is there any specific reason why 100%, 50%, and 0% SOC were chosen for testing? Do these SOC's correspond to anything in particular?
3. Are the placement and operation of the heater chosen to replicate a specific situation that is likely to be seen during normal cell operation, or is the intention just to increase the temperature of the cell under test until it reaches thermal runaway?
4. Is there any variability in the amount of charge/discharge cycles the cells experience prior to the testing? More cycles would likely cause the SEI layer to be thicker and result in more degradation and the possibility of lithium dendrite formation, which could all affect the response of cells when they reach thermal runaway.
5. During sensor insertion, is the cell contained in a glovebox or is it exposed to atmosphere? The exposure of internal cell components to oxygen and water vapor can lead to oxidation and increased degradation, which could affect the long-term operation of the cells after the sensors are inserted.

Improvements:

- The method of using the FBG/FPI sensors in batteries could add be an effective tool in the battery research. The claim of proposing the sensor for the first time is however not novel. For instance, <https://doi.org/10.1364/AO.53.002136> used the FBG/FBI sensor in monitoring the thermal ablation in the liver tumors. They also suggested the decoupling the pressure and temperature measurement by measuring the peak wavelength shift and air-gap compression. Another example of FBG - FPI based sensors proposed by <https://doi.org/10.3390/app10031069> in structural health monitoring where they measured the sensitivity of theses sensor both towards temperature and strains. They also highlighted the quality of these sensors being small, unaffected from electromagnetic interference as key aspects. So, the claims should be limited to "use" of these sensors in monitoring the thermal runaway of batteries and predecessors should be given the due credit.
- The geometrical depiction of the FBG and FPI in picture 1 is unclear. The lengths of the FBG and FPI,

as well as the space between them, are incorrectly represented. The size and dimension are missing, and the box zoom in Figures 1B and E has the same origin. Figure 1 b mixes 2D and 3D graphics.

- Because the temperature was also measured at several points using thermocouples, an error assessment with the temperature predicted by the FBG is needed.
- It would be an exaggeration to suggest that the association between thermal runaway and optical responses is "precisely quantified" because the pressure rate and temperature rate curves are continuous in nature and there is no sharp yielding detected. Instead, a range should be specified between the warning stage and irreversible damage.

Reviewer #2 (Remarks to the Author):

This manuscript approaches the discrimination and tracking of temperature and pressure in thermal runaways' situations of LiBs by optical fiber sensors. The English language is ok (minor typos in the introduction and abstract sections). Overall, a very good and complete description of the work developed. However, the pressure and temperature discrimination performed by this type of sensing technology inside of LiBs is not a novelty work. The optical sensor design is also not novel. This work will be significant in the field of improved battery management systems. However, there are some important references missing. Similar works using a similar sensing approach were already reported in the literature and are not referenced in this manuscript:

- [1] doi.org/10.1016/j.jpowsour.2018.10.096;
- [2] doi.org/10.1016/j.measurement.2022.111961;
- [3] doi.org/10.1016/j.est.2022.105260;

1- In the Conclusion section, the authors state that "This photonics-based sensing technology opens up previously unavailable opportunities for ensuring battery safety. Above all, the distributed optical fiber sensors can provide access to the possibility of monitoring numerous points simultaneously using the same optical fiber". Can the authors please comment on the meaning of this sentence, regarding the optical fiber distributed sensing approach? There is no discussion in the rest of the manuscript regarding this approach. How do the Authors propose to perform and interrogate different FPI sensors just in one fiber line? Or is this related to Brillouin or Raman distributed sensing?

2- In recent years, the use of fiber optic sensors for battery applications has been largely reported. However, the high cost of the interrogation system, e.g., the interrogator and optical spectrum analyser used in the present work, has hindered the adoption of fiber sensors in real scenarios. What is the authors' vision on this aspect?

3- Sentences repetition on the figure's description along the main text and in the figure description. Please revise this aspect.

4- FPI fabrication: which manual splice parameters were used? Please provide details regarding the fusion splicer and a comparison with a standard splice.

5- Figure 4: Internal T values are lower than the ones at the surface. Why?

6- Figure 5: kPa in yy axis. Should it be in MPa?

7- Why did not the Authors use the same interrogation system in the calibration process and during the experimental tests? Will this approach have any impact on the sensor's calibration? More details regarding the interrogation setup experimental resolutions and spectral range should also be provided.

8- References standardization is missing.

9- Similar figures were identified in the Supplementary information document, S6-b (doi.org/10.1016/j.measurement.2022.111961) and S7-c,d,e,f (doi.org/10.1364/OE.22.021757) without any references. It is assumed that one manuscript submitted to a very prestigious journal such as Nature Communications should bring excellent and new inputs to the scientific community beyond the state of the art, and this is not verified in this case.

10- The tube protection diameter used for the fiber optic sensor's instrumentation on the LiB significantly increases their invasiveness. How could this approach impact the results regarding the pressure and temperature flow inside the battery?

11- The optical sensing approach used to pressure detection in the manuscript is based on one open Fabry-Perot interferometer. However, in some of the battery applications (EVs) different LiB formats are used. Can you please comment on how this type of sensor can be used for a different LiB configuration, such as the pouch cell or coin cell, and please comment about their battery instrumentation regarding their handling and fragility (due to the C-shape fiber format used)? Is this feasible for a mass manufacturing scale?

12- Almost all the methods reported are ok. However, the authors did not comment on why a mixture of just 3% for each different gas concentration was used (Figure S3). There are already some works reported in the literature (e.g. doi.org/10.1364/OL.37.003063) where it is shown that the sensitivity of open FPI sensors changes in presence of different gases (due to refractive index variations, which are different for each gas).

13- Also, the deduction performed from equation 4 does not seem to be correct. There are several studies in the literature regarding the pressure variation with open cavities in air related to the refractive index shift. The nm parameter is not equal to P values. Therefore, the P value in Equation 5 is not correct. The authors are advised to check the following references and correct this Section. doi.org/10.1109/JPHOT.2015.2489926; doi.org/10.1364/OE.23.023484.

14- The equation used to describe the FPI interference condition is not correct (please see page 3, 1st paragraph, and in Figure 1f). As it is, the equation defines the free spectral range.

15- During the fabrication of the FBG-FPI, the authors did not clarify which part of the sensor is created first. On page 9, the authors mention that "The cut open-cavity fiber is then spliced to another SMF using manual alignment, and the other end is pinched off directly to avoid unwanted reflections". However, the FBG sensors seem to be created after the FPI sensors. What is the distance between the FPI and fiber end? What is the impact of having a low reflectivity FPI before/after the FBG sensor (in relation to the interrogation system)? More details should be provided.

16- In the FPI production, what is the uncertainty associated with the fabrication method?

Thus, the manuscript in its present state and whit all the early points mentioned does not present the novelty and quality for being published in the prestigious Nature Communications Journal. It is my opinion that the manuscript should be rejected for publication in this Journal.

Author's Response to the Reviewers

Title: Operando monitoring of thermal runaway in commercial lithium-ion cells via advanced lab-on-fiber technologies

April 17, 2023

Dear Reviewers,

We have considered all the questions and suggestions from you and revised our manuscript accordingly. Below please find point-by-point responses to your comments and the changes made in the revised version (underlined and highlighted in red colour).

To facilitate reading of this letter, we have, (i) systematically reported your questions or comments in italics followed by our response, and (ii) handled each referee separately. Besides, we have redrawn the figures that overlap with our previous work as requested.

Meanwhile, we made three additional changes in the revision. We hope these changes can be accepted.

1. We added one more author (Dr. Xiaobin Xue) and reordered parts of the authors in the author list as they carried out new experiments required.
2. We provided the information of "Competing interests" and "Author contributions" in the last part of the revised main article.
3. Consider the references in our original version are too much (over 60 references), we removed parts of overlapped ones with similar contributions (now 49 references).

Having done our best to satisfactorily answer your comments and requests for extra data, we hope that this revised version could meet the high standards of *Nature Communications*.

Yours sincerely,

Reviewer #1:

The manuscript discusses the use of compact fiber optic sensors inside of commercial 18650 Li-ion batteries for monitoring internal temperatures and pressures during thermal runaway. This allows for more direct and accurate monitoring of cell characteristics during testing, and can provide early indication of thermal runaway in cells during operation. The authors give a mostly thorough explanation of the sensor insertion and subsequent testing, as well as analysis of results.

The authors employed the fiber optics sensors Fabry-perot interferometer (FPI) in combination with the fiber bragg gating (FBG) for operando monitoring of the cell thermal runaway reactions in cylindrical lithium ion phosphate batteries. The single sensor is used to monitor the temperature and pressure during the thermal runaway using the Bragg wavelength variation. The pressure change is then decoupled using the peak wavelength shift of the interferometric spectrum of the FPI which is regarded as a function of pressure only. Detecting the turning sections of the time derivative curves of the pressure and temperatures are used in establishing a safety warning signal. The study would help in the battery health monitoring and generating an alarming signal much before the irreversible damage takes place.

However, the manuscript would be improved by addressing the following comments.

Comments/questions:

[1] What is the manufacturer-rated lifetime of the LFP cells used in the testing? Normally LFP cells are expected to last hundreds or sometimes thousands of cycles, but the capacity fade in Figure 2 (f) shows the cells lose ~20% of their capacity in only 100 cycles. It would be helpful to have a table that lists the cell specifications provided by the manufacturer.

Response: This is an important concern should be clarified. The charge-discharge rate recommended by the cell manufacturer is normally “0.5 C-rate” (i.e. applied current equals to 0.5*rated current), and the manufacturer-rated lifetime of commercial cell product specification is defined as “discharge capacity (1000th Cycle) > 80% SOH at 0.5 C-rate”. Different from it, in Figure 2 (f) we used 2 C-rate for comparison to achieve a rapid capacity degradation. As we stated in the Methods “The charge rate of 2 C is chosen because once the unaffected cell performance is guaranteed at high rates, it will be fine at lower rates”. The rapid capacity degradation at high rate is mainly ascribed to the SEI thickening and Li plating on anode graphite that induces the loss of active Li inventory [R1,R2]. In order to clarify above concern, we have added Table S1 in Supplementary Information to list the specifications of cell, involving rated capacity, size, weight, voltage range, internal resistance, rated voltage and cycle life.

Manuscript modifications: The discussions above and the cell specifications provided by the manufacturer have been added in the revised manuscript (see the red colored texts in Methods, page 10), and the supplementary information (see the red colored Table S1, page 2).

[R1] Tomaszewska, A., Chu, Z., Feng, X., O’Kane, S., Liu, X., Chen, J., Ji, C., Endler, E., Li, R., Liu, L., Li, Y., Zheng, S., Vetterlein, S., Gao, M., Du, J., Parkes, M., Ouyang, M., Marinescu, M., Offer, G. & Wu, B. Lithium-ion battery fast charging: A review. *eTransportation* 1, 100011 (2019).
[R2] Birkel, C. R., Roberts, M. R., McTurk, E., Bruce, P. G. & Howey, D. A. Degradation diagnostics for lithium ion cells. *J. Power Sources* 341, 373-386 (2017).

[2] Is there any specific reason why 100%, 50%, and 0% SOC were chosen for testing? Do these SOC's correspond to anything in particular?

Response: As we know, the cell's thermal runaway behaviors and characteristics (such as trigger conditions, safety venting, voltage drop, maximum temperature, maximum pressure, etc.) are different at different SOC's. Therefore, it is necessary to select representative SOC's to study the inherent rule of cell thermal runaway.

1) The selection of 100%, 50%, and 0% SOC's is a general consensus among researchers for thermal runaway studies, references [R3-R7] all select the three SOC's for comparison. Therefore, our studies here is not aimed at a particular case.

2) The 100% SOC indicates the full-charged battery with the largest electric energy stored inside the cell, which can simulate the most severe condition in actual situation that is nearly covered in all thermal runaway researches. On the contrary, although the battery with 0% SOC represents little electric energy stored inside the cell, it does not mean absolute safety because of the occurrence of SEI decomposition, easily melted separator, anode/cathode reaction, and flammable electrolyte, the thermal runaway of the battery with 0% SOC is also necessary. Finally, 50% SOC is recommended for battery storage conditions [R8] that is necessary to study thermal runaway behavior, and 50% SOC is also between 0% SOC and 100% SOC for better comparison.

[R3] Ribi re, P., Grugeon, S., Morcrette, M., Boyanov, S., Laruelle, S. & Marlair, G. Investigation on the fire-induced hazards of Li-ion battery cells by fire calorimetry. *Energ. Environ. Sci.* 5, 5271-5280 (2012).

[R4] Liu, P., Li, Y., Mao, B., Chen, M., Huang, Z. & Wang, Q. Experimental study on thermal runaway and fire behaviors of large format lithium iron phosphate battery. *Applied Thermal Engineering* 192, 116949 (2021).

[R5] Kvasha, A., Guti rrez, C., Osa, U., de Meatz, I., Blazquez, J. A., Macicior, H. & Urdampilleta, I. A comparative study of thermal runaway of commercial lithium ion cells. *Energy* 159, 547-557 (2018).

[R6] Friesen, A., Horsthemke, F., M nnighoff, X., Brunklaus, G., Krafft, R., B rner, M., Risthaus, T., Winter, M. & Schappacher, F. M. Impact of cycling at low temperatures on the safety behavior of 18650-type lithium ion cells: Combined study of mechanical and thermal abuse testing accompanied by post-mortem analysis. *J. Power Sources* 334, 1-11 (2016).

[R7] Galushkin, N. E., Yazvinskaya, N. N. & Galushkin, D. N. Mechanism of Thermal Runaway in Lithium-Ion Cells. *J. Electrochem. Soc.* 165, A1303 (2018).

[R8] Fang, J., Cai, J. & He, X. Experimental study on the vertical thermal runaway propagation in cylindrical Lithium-ion batteries: Effects of spacing and state of charge. *Applied Thermal Engineering* 197, 117399 (2021).

[3] Are the placement and operation of the heater chosen to replicate a specific situation that is likely to be seen during normal cell operation, or is the intention just to increase the temperature of the cell under test until it reaches thermal runaway?

Response: Yes, our idea of heater used here is similar to adjacent two cells in a cell modules or pack. The battery cells are arranged in modules to achieve serviceable units. The cells are connected in series and in parallel, into battery packs, to achieve the desired voltage and energy capacity.

In cell module or pack, the cells are closed attached to each other. Therefore, the operation of the cylindrical heater used here is just like a cell has experienced thermal runaway and to affect its adjacent cells in the module. The cylindrical heater is with the same size as the adjacent cell. Therefore, our heating mode will be helpful for the thermal diffusion and heat propagation in cell-module-pack system, as shown in Figure 1, which will be significantly meaningful for safety studies of energy storage stations and electric vehicles [R9, R10].

Figure 1 The heater used for aimed at study of heat propagation in cell-module-pack system.

[R9] Zhang, Q., Niu, J., Zhao, Z. & Wang, Q. Research on the effect of thermal runaway gas components and explosion limits of lithium-ion batteries under different charge states. *Journal of Energy Storage* 45, 103759 (2022).

[R10] Zhong, G., Li, H., Wang, C., Xu, K. & Wang, Q. Experimental Analysis of Thermal Runaway Propagation Risk within 18650 Lithium-Ion Battery Modules. *J. Electrochem. Soc.* 165, A1925 (2018).

Manuscript modifications: The discussions above and the reason for the heating mode have been added in the revised manuscript (see the red colored texts in page 4).

[4] Is there any variability in the amount of charge/discharge cycles the cells experience prior to the testing? More cycles would likely cause the SEI layer to be thicker and result in more degradation and the possibility of lithium dendrite formation, which could all affect the response of cells when they reach thermal runaway.

Response: This is very important issue. Based on our experimental results, there is no variability in the amount of charge/discharge cycles the cells experience prior to the testing. Before thermal runaway testing, the fresh 18650 LFP cells are pre-cycled for three times for SEI

formation (this procedure is a consensus before testing). The discharged cells (0% SOC) are prepared for drilling to reduce the risk during drilling. The modified cells are firstly charged to 100% SOC and then discharged to the prescribed SOC (50%, 0% SOC) for thermal runaway tests to ensure the performance of each modified cell. No excessive cycling is performed before thermal runaway tests.

The cycle times can indeed influence the thermal runaway behavior due to the thickened SEI layer and Li dendrite formation (if fast charging). Our previous studies and other researches have been focused on the aging effect on thermal runaway external characteristics. However, the thermal runaway behavior varies with the degradation path and mechanism that manifests as different response as the decline of state of health (SOH, ratio of current capacity to initial capacity). We have summarized two cases in Figure 2, as below:

Case 1: If more Li dendrites are formed after cycling during fast charging, low temperature charging or overcharging, the battery can manifest as declined thermal stability (easily to trigger thermal runaway) due to the violent exothermic reaction between electrolyte and highly active Li metal [R11] (Figure 2a).

Case 2: When the SEI thickening induces loss of Li inventory, which causes reduced lithiation degree of the fully-charged anode after aging (such as after high-temperature storage), resulting in delayed onset temperature of thermal runaway and reduced heat generation, the thermal stability may have a little improvement or display no obvious change [R12] (Figure 2b).

Figure 2 The effects of battery degradation on thermal runaway behavior [R11, R12].

This study is mainly focused on the internal temperature-pressure monitoring at different thermal runaway behavior and highlights the early warning signal of thermal runaway. We do find that the internal battery state evolution during thermal runaway upon capacity degradation is complex and it is affected by multiple factors (such as degradation paths, aging mechanism and battery chemistries, etc.). This is really a challenging topic and we will work on it in-depth study!

[R11] Friesen, A., Horsthemke, F., Mönninghoff, X., Bruncklaus, G., Krafft, R., Börner, M., Risthaus, T., Winter, M. & Schappacher, F. M. Impact of cycling at low temperatures on the safety behavior of 18650-type lithium ion cells: Combined study of mechanical and thermal abuse testing accompanied by post-mortem analysis. *J. Power Sources* 334, 1-11 (2016).

[R12] Ren, D., Hsu, H., Li, R., Feng, X., Guo, D., Han, X., Lu, L., He, X., Gao, S., Hou, J., Li, Y., Wang, Y. & Ouyang, M. A comparative investigation of aging effects on thermal runaway behavior of lithium-ion batteries. *eTransportation* 2, 100034 (2019).

[5] During sensor insertion, is the cell contained in a glovebox or is it exposed to atmosphere? The exposure of internal cell components to oxygen and water vapor can lead to oxidation and increased degradation, which could affect the long-term operation of the cells after the sensors are inserted.

Response: Yes, the implanting processes are implemented in the glovebox. We have added some pictures to demonstrate the implanting steps in glove box, as shown in Figure 3.

Figure 3 Cell modification and sensor implanting processes inside the glovebox.

Manuscript modifications: The discussions above have been added in the revised manuscript (see the red colored texts in Methods, page 10) and the supplementary information (see the red colored Fig. S5, page 7)

Improvements:

[6] The method of using the FBG/FPI sensors in batteries could add be an effective tool in the battery research. The claim of proposing the sensor for the first time is however not novel. For instance, <https://doi.org/10.1364/AO.53.002136> used the FBG/FPI sensor in monitoring the thermal ablation in the liver tumors. They also suggested the decoupling the pressure and temperature measurement by measuring the peak wavelength shift and air-gap compression.

Another example of FBG - FPI based sensors proposed by <https://doi.org/10.3390/app10031069> in structural health monitoring where they measured the sensitivity of these sensor both towards temperature and strains. They also highlighted the quality of these sensors being small, unaffected from electromagnetic interference as key aspects. So, the claims should be limited to "use" of these sensors in monitoring the thermal runaway of batteries and predecessors should be given the due credit.

Response: Many thanks for the Reviewer's great reminder and suggestion. We do appreciate the pioneering work and outstanding contributions of researchers on FBG/FPI and related optical fiber sensing technologies. We have claimed the novelty of above studies in the Introduction and added them in the References, to express our great appreciation!

Base above pioneering works, we further achieve the following progresses in battery monitoring: **1) Excellent sensor's characteristics under hazard conditions:** we proposed and fabricated a cascade sensor with a femtosecond-laser-inscribed FBG and an open cavity FPI, the sensor's performance is excellent at high temperature and pressure environments, offering a highly linear response, very limited crosstalk and good repeatability before and after thermal runaway. **2) Advanced signal analysis for cell safety early warning:** scalable solution for predicting imminent thermal runaway is found by the detection of the abrupt change of slope of the differential curves of cell temperature and pressure, which corresponding an internal transform between cell's reversible and irreversible reactions.

Manuscript modifications: The discussions above have been added in the revised manuscript (see the red colored texts in Introduction, page 1-2 and Refs. 15, 16).

[7] The geometrical depiction of the FBG and FPI in picture 1 is unclear. The lengths of the FBG and FPI, as well as the space between them, are incorrectly represented. The size and dimension are missing, and the box zoom in Figures 1B and E has the same origin. Figure 1 b mixes 2D and 3D graphics.

Response: We have redrawn the Fig.1 as below. We hope this new version will show a clear geometrical depiction, operation principle and spectral character of optical fiber sensor implanted inside the cell. The sensing probe is assembled from two sections, one is a FBG (0.8 mm in grating length) and the other is a FPI (0.1 mm in open cavity length). There is a 10 mm fiber gap between them to avoid optical interference. The resulting total length of the sensor is about 12 mm with a 125 μm uniform diameter. The sensing probe is spliced at the end of a standard communication single-mode optical fiber and located into the central hole of the cell.

Figure 4 FBG & FPI sensor for simultaneous temperature and pressure monitoring in the cell.

Manuscript modification: The Fig.1 have been redrawn in the revised manuscript to show clear geometrical depiction of FBG and FPI (see the revised Fig.1 on page 2).

[8] Because the temperature was also measured at several points using thermocouples, an error assessment with the temperature predicted by the FBG is needed.

Response: Yes, there are several temperature measurement points (internal cell, cell surface and heater) collected by several thermocouples. However, the key point is still inside the cell, i.e. the internal temperature error between FBG and thermocouple at the same position in the central hole of the cell.

We use the concept of “Pearson correlation coefficient” to better address this concern and carried out additional experiments to achieve a comprehensive error assessment. As shown in Figure 5 (a), both FBG and thermocouple (TC) are implanted at the same position into the 18650 cells. The Pearson correlation coefficient (*PCC*) [R13, R14] used here is used to quantitatively verify the linear relationship between the temperature of FBG and TC, and to present the absolute and relative error of the maximum temperature between them. The *PCC* are shown in Equation (1) [R15],

$$PCC = \frac{\sum_{i=1}^n (T_{FBG_i} - \overline{T_{FBG}})(T_{TC_i} - \overline{T_{TC}})}{\sqrt{\sum_{i=1}^n (T_{FBG_i} - \overline{T_{FBG}})^2} \sqrt{\sum_{i=1}^n (T_{TC_i} - \overline{T_{TC}})^2}} \quad (1)$$

where T_{FBG} and T_{TC} represent the temperature measured by FBG and TC, respectively. *PCC* value is between -1 and 1, where 1 or -1 represents a 100% linear relevance, and 0 means 0% relevance.

1) Error assessment during normal cycling: Firstly, a consecutive charge-discharge cycling at 0.5 C, 1 C, 1.5 C and 2 C is performed with 30-minute relaxation setting between charge and discharge for temperature recovery. The internal temperature evolution of the FBG and TC is nearly identical as shown in Figure 5 (b). The Pearson correlation coefficient is 99.86%,

the absolute error between the maximum temperature measured by FBG and TC during normal cycling is 0.12 °C, the relative error is merely 0.31% as shown in Figure 5 (d), displaying the neglect error during battery normal cycling.

2) Error assessment during thermal runaway: The battery with 100% SOC is taken representatively to complete the error assessment, the cell is triggered thermal runaway by overheating of a 100 W heater (the same condition in the main text). The internal temperature monitored by FBG and TC is also highly overlapped as shown in Figure 5 (c). The Pearson correlation coefficient is 99.89%, the absolute error between the maximum temperature measured by FBG and TC during normal cycling is 2.43 °C, the relative error is merely 0.46% as shown in Figure 5 (d), displaying the neglect error during battery thermal runaway.

Figure 5 Comprehensive temperature error assessment between thermocouple and FBG.

[R13] Zhou, Y., Huang, M., Chen, Y. & Tao, Y. A novel health indicator for on-line lithium-ion batteries remaining useful life prediction. *J. Power Sources* 321, 1-10 (2016).

[R14] Ma, Y., Shan, C., Gao, J. & Chen, H. A novel method for state of health estimation of lithium-ion batteries based on improved LSTM and health indicators extraction. *Energy* 251, 123973 (2022).

[R15] Conover, W. J. *Practical nonparametric statistics*. Vol. 350 (John Wiley & Sons, 1999).

Manuscript modifications: The error assessment methods have been added in the revised manuscript (see the red colored texts in page 3 and Methods, page 10). The Figure 5 has been added in the supplementary information (see the red colored Fig. S3, page 4-5)

[9] It would be an exaggeration to suggest that the association between thermal runaway and optical responses is "precisely quantified" because the pressure rate and temperature rate curves are continuous in nature and there is no sharp yielding detected. Instead, a range should be specified between the warning stage and irreversible damage.

Response: Many thanks for the Reviewer's valuable suggestion. I do agree this your idea! It is true that the experimental achieved "turning point" of temperature and pressure rate is not sharp and it should be expressed as a "turning range" (also well agree with the cell model analysis). In order to correctly describe this key idea and make it easier to understand, we used the expression of "warning range", which is starting from the electrolyte evaporation and ending at the SEI decomposition. The corresponding temperature range is from 70 °C to 80 °C. The safety early warning range is setting by detection of switch from reversible physical change and irreversible chemical reaction, which guarantees the safety use of cell before irreversible damage.

Figure 6 Setting a safety warning by finding of the "switch range" between reversible and irreversible reactions via calculation the temperature and pressure time derivatives.

Manuscript modifications: The discussions above have been added in the revised manuscript (see the red colored texts in page 6-7). The Abstract has been revised accordingly.

Reviewer #2:

This manuscript approaches the discrimination and tracking of temperature and pressure in thermal runaways' situations of LiBs by optical fiber sensors. The English language is ok (minor typos in the introduction and abstract sections). Overall, a very good and complete description of the work developed. However, the pressure and temperature discrimination performed by this type of sensing technology inside of LiBs is not a novelty work. The optical sensor design is also not novel. This work will be significant in the field of improved battery management systems. However, there are some important references missing. Similar works using a similar sensing approach were already reported in the literature and are not referenced in this manuscript:

[1] doi.org/10.1016/j.jpowsour.2018.10.096;

[2] doi.org/10.1016/j.measurement.2022.111961;

[3] doi.org/10.1016/j.est.2022.105260

Response: We appreciate the reviewer capturing the underlying science and the practical impact of the work. And we are sorry that so many important references are not cited. This is a non-negligible mistake! Above mentioned references made pioneering and outstanding contributions to *operando* monitoring of batteries and their performance analysis.

As the reviewer comments, the key point for this work is it focused on battery thermal runaway and will be significant in the field of improved battery management systems. Our work is learning from above pioneering references, and making following two progresses in battery monitoring and safety warning:

1) Excellent sensor's characteristics under hazard conditions: we proposed and fabricated a cascade sensor with a femtosecond-laser-inscribed FBG and an open cavity FPI, the sensor's performance is excellent at high temperature and pressure environments, offering a highly linear response, very limited crosstalk and good reproducibility before and after thermal runaway. The physical-chemical-electrochemical reactions in cell will be much complexity during thermal runaway accompanied by the high temperature, high pressure, gases and smoke release, even combustion and explosion, as shown in Figure 7. Therefore, most of the sensors cannot survive after thermal runaway, not to say a stable and reproducible output with limited crosstalk.

Figure 7 Different conditions between battery normal cycling and thermal runaway [R16, R17]

2) Advanced signal analysis for cell safety early warning: establishing a safety warning range by detection of the switch range from reversible and irreversible reaction through temperature and pressure time derivatives, is another key contribution of this work which will be meaningful in emerging renewable energy relevant applications for many substances that require detection and quantification, for example the energy storage stations and the electric vehicles. It is the true multidisciplinary efforts between photonics and electro-chemistry groups, together with data science research, can we have led to above impressive progresses.

Figure 8 A timeline involving the representative references towards internal temperature, strain and pressure monitoring of lithium-ion battery.

In summary, we plot a timeline involving the references the reviewer and editor proposed, as well as some representative references towards the internal characteristics monitoring of lithium-ion battery as shown in Figure 8. It can be seen that, over the past ten years, great efforts have been made in internal temperature, strain, and pressure monitoring of the lithium ion battery during normal cycling via optical fiber sensors, while no attempt have been made towards battery thermal runaway monitoring. Our current study thereby addresses this important topic. By raising an alert even before safety venting, our sensor represents an urgently needed breakthrough in cell safety assessment and warning of thermal runaway.

[R16] Wang, Q. S., Jiang, L. H., Yu, Y. & Sun, J. H. Progress of enhancing the safety of lithium ion battery from the electrolyte aspect. *Nano Energy* 55, 93-114 (2019).

[R17] Mei, W., Zhang, L., Sun, J. & Wang, Q. Experimental and numerical methods to investigate the overcharge caused lithium plating for lithium ion battery. *Energy Storage Mater.* 32, 91-104 (2020).

Manuscript modifications: The discussions above have been added in the revised manuscript (see the red colored texts in Introduction, page 1, Refs. 12, 20).

[1] In the Conclusion section, the authors state that “This photonics - based sensing technology opens up previously unavailable opportunities for ensuring battery safety. Above all, the distributed optical fiber sensors can provide access to the possibility of monitoring numerous points simultaneously using the same optical fiber”. Can the authors please comment on the meaning of this sentence, regarding the optical fiber distributed sensing approach? There is no discussion in the rest of the manuscript regarding this approach. How do the Authors propose to perform and interrogate different FPI sensors just in one fiber line? Or is this related to Brillouin or Raman distributed sensing?

Response: We are sorry we make a mistake on this expression. It is true that FPI sensors are hard to be multiplexed in one fiber (potentially can be achieved but not fit for in operando battery monitoring). Here in the conclusion, we hope to give a prospect on the future development of optical fiber sensing technologies for battery research and applications.

Manuscript modifications: The following two paragraphs have been added in the final part of the Conclusion (see the red colored texts in Conclusion, page 8-9).

“With the development of clean-energy systems, ranging from electric cars to power plants, it is meaningful to realize a portable and cost-effective interrogator for in field measurement. By using a tunable laser as a source (for example, a compact VCSEL48 together with a photodiode as detector and an analog-to-digital converter (A/D) to obtain the desired data, traditional broadband light sources and expensive optical spectrum analyzers can be replaced. The function of the tunable laser is to match the wavelength of the most sensitive spectral region of the FPI and FBG, so once that the sensor is characterized it can be replaced by a common laser. This technique relies on the principle of edge filtering so that the detected optical power changes as a result of a wavelength shift of the mode resonance with respect to the fixed wavelength of the laser source (see Fig. S16).

Last but not least, true multidisciplinary efforts between photonics, material and electro-chemistry groups have led to impressive progress in emerging renewable energy relevant applications for many substances that require detection and quantification⁴⁹. It is these efforts that are allowing the full potentials of advanced lab-on-fiber technologies to be reached. Various key parameters including temperature, pressure, refractive index, gas and ions can be monitored simultaneously in operando over one optical fiber, at multi-positions of the battery. This provides theretofore unrealizable crucial capabilities in safety of operation as well as complementary information regarding battery state of health and evolution. Given the potential for optoelectronic integration of the components needed, it is not unthinkable to envision widespread use of these techniques in mass market applications, such as electric vehicles.”

[2] In recent years, the use of fiber optic sensors for battery applications has been largely reported. However, the high cost of the interrogation system, e.g., the interrogator and optical spectrum analyser used in the present work, has hindered the adoption of fiber sensors in real scenarios. What is the authors' vision on this aspect?

Response: Yes! This is a very important question we have asked ourselves for years. Many thanks for the Reviewer pointed it out. To realize a portable instrumentation for in-field measurement, we can use a real-time interrogation scheme by replacing the wavelength interrogation by optical power detection. In this case, a tunable laser (TLS) can be used as a source instead of BBS, together with a photodiode as detector and an analog-to-digital converter

to obtain the desired data (to replace the OSA). The function of the TLS is matching the wavelength of the most sensitive Bragg and Fabry-Perot interference modes, so once that the sensor is characterized it can be replaced by a common laser (for example, a compact VCSEL). This technique relies on the principle of edge filtering so that the optical power change is produced due to the wavelength shift of the mode with respect to the fixed wavelength of the laser source. Our lab made a prototype of the instrumentation using above power detection principle, see the Figure 9. Such compact interrogator has been reported in several groups [R18, R19, R20].

Figure 9 (a) Present lab-based spectral interrogation system and (b) portable power detection instrumentation for in-field measurement.

[R18] Y. H. Huang, T. Guo, C. Lu, H. Y. Tam, "VCSEL-based tilted fiber grating vibration sensing system," IEEE Photon. Technol. Lett. 22, 1235-1237 (2010).

[R19] S. Ura, S. Shoda, K. Nishio, Y. Awatsuji, In-line rotation sensor based on VCSEL behavior under polarization-rotating optical feedback, Opt. Express 19, 23683-23688 (2011).

[R20] T. Guo, F. Liu, F. Du, Z. Zhang, C. Li, B. Guan, J. Albert, VCSEL-powered and polarization-maintaining fiber-optic grating vector rotation sensor, Opt. Express 21, 19097-19102 (2013).

Manuscript modifications: The portable power detection method has been added in the revised manuscript (see the Conclusion, page 8) and the supplementary information (see the Figure S16, page 19).

[3] Sentences repetition on the figure's description along the main text and in the figure description. Please revise this aspect.

Response: Thanks for the careful observation, we have revised the figure's description and the main text, please check them.

[4] FPI fabrication: which manual splice parameters were used? Please provide details regarding the fusion splicer and a comparison with a standard splice.

Response: Both standard splicing mode and manual mode can achieve FPI fabrication, see the details of Table 1.

Table 1 Detailed information on standard splice parameters and manual splice parameters.

Splicing type	Splicing mode	Discharge power	Discharge time	Discharge position
Standard splice	Auto SM	82 bit	2200 ms	0 um near the end of the SMF
Manual splice	Normal SM	26 bit	2300 ms	15 um near the end of the SMF
JILONG fusion splicer, KL-380				

Standard splicing: power "standard" is the default power of KL-380, JILONG fusion splicer for splicing single-mode fiber, the splicing mode is Auto SM mode, the fiber splicing discharge strength is 82 bit, time is 2200 ms, as shown in Figure 10a.

Manual splicing: fusion splice mode is Normal SM mode, fiber splicing discharge intensity is 26 bit, time is 2300 ms. Place the cut single-mode fiber and open cavity fiber into the fusion splicer, select core alignment mode, the fusion splicer is automatically aligned, and then moves through the drive motors (ZL and ZR) on the fusion splicer, also in the X and Y planes. Select the Adjust Fiber button to discharge the fusion splice at 15 um near the end of the single-mode fiber, as shown in Figure 10b.

Figure 10 Diagram of manual splice discharge.

[5] Figure 4: Internal T values are lower than the ones at the surface. Why?

Response: This is because heating is from the outside of the battery. Therefore, at the first stage, the internal temperature is lower than the surface before thermal runaway. While internal temperature will exceed the surface temperature gradually and much higher than the surface when the cell reaches to thermal runaway. According to the heat transfer theory, the battery temperature depends on both heat generation and heat dissipation, we divide three stages to explain the internal and surface temperature evolution as shown in Figure 11. The specific illustrations are as follows:

Stage 1: At the early stage of heating (before side reaction): there has no side reaction inside the cell, at which the battery can be regarded as an anisotropic heat conductor, the increase in cell temperature comes from the heat conduction of the heater, following Fourier's law (Equation (2)). The thermal conductivity along radial direction (λ_r) is lower than that along the axial direction (λ_a) [R21], the lower thermal conductivity slows down the heat conduction towards the center of battery, resulting in lower internal temperature than the surface temperature.

$$J_T = -\lambda \frac{dT}{dx} \quad (2)$$

where J_T indicates heat flux, λ is thermal conductivity.

Stage 2: Onset of side reactions until thermal runaway: as the temperature of the battery increases, the side reactions begin with SEI decomposition to facilitate increase of temperature. The increase in cell temperature comes from both the heat conduction of the heater and the internal side reactions (includes SEI decomposition, separator melting and internal short circuit), where the heat conduction of the heater dominates the total heat sources of the battery (because the internal side reaction heat generation at this stage is small from the DSC curves as shown in Fig. S12), also leading to the lower internal temperature than the surface temperature. However, the internal short circuit leads to transient Joule heating, causing a temporary internal temperature jump (the zoomed inset of Fig. 4a in the manuscript), and then recover back to the initial increase rate.

Stage 3: Thermal runaway: when the cell triggers thermal runaway, the heater is stopped, the internal temperature exceeds surface temperature ascribed to the predominant and transient side reaction heat generation inside the battery that spread from internal to the surface and cannot rapidly dissipate, leading to the battery internal temperature to exceeds the heater temperature and battery surface temperature as displayed in Fig.3d in the manuscript.

Figure 11 The internal and surface temperature evolution at three stages

[R21] Chen, S. C., Wan, C. C. & Wang, Y. Y. Thermal analysis of lithium-ion batteries. *J. Power Sources* 140, 111-124 (2005).

Manuscript modifications: The discussions above have been added in the revised manuscript (see the red colored texts in page 6, Ref. 39).

[6] Figure 5: kPa in yy axis. Should it be in MPa?

Response: We have double checked the Fig. 5 and made sure that the unit should be kPa (not similar with the unit shown in Fig. 4). This is because the pressure changes shown in Fig. 5 is much SMALLER that Fig. 4. What we carefully studied in Figs.5 (c, d, e) are the zoomed part (marked with a red frame) in the Fig.5 (b), where the pressure change is very slight. It is because the sensor offers a high sensitivity, real time and very stable response, we may find this important information.

[7] Why did not the Authors use the same interrogation system in the calibration process and during the experimental tests? Will this approach have any impact on the sensor's calibration? More details regarding the interrogation setup experimental resolutions and spectral range should also be provided.

Response: We appreciate the reviewer's so careful question. Yes, we used two kinds of spectral interrogators in this work. One is the lab-based optical spectrum analyzer (AQ6380, YOKOGAWA), another is the industry-based optical spectral interrogator (FS22SI, HBM).

The reason we use above two spectral interrogators fit for different experimental environments. Optical spectrum analyzer (AQ6380, YOKOGAWA) is used in a clean lab for optical fiber sensor fabrication, calibration and packaging. All these tasks are performed at Institute of Photonics Technology, Jinan University. On the other hand, the optical spectral interrogator (FS22SI, HBM) is used for battery thermal runaway measurement. We may simultaneously measure several batteries and real time readout the data over the whole testing process. All these works are carried out at the State Key Laboratory of Fire Science, University of Science and Technology of China.

The optical spectrum analyzer (AQ6380, YOKOGAWA) takes 16 s to acquire the full spectrum (range 1500-1600 nm) of an optical fiber sensor at 1 pm resolution and HIGH1 mode. In the actual thermal runaway monitoring process, this may lead to inaccurate acquisition of temperature and pressure information due to the relatively fast thermal runaway time of the battery. The optical spectral interrogator (FS22SI, HBM) can acquire the full spectrum (range 1500-1600 nm) of the optical fiber sensor at a refresh rate of 1 s, and simultaneously acquiring 8 channel data for multi batteries thermal runaway monitoring.

Table 2 Detailed setup parameters for different interrogation system.

Instrumentations	Experimental labs	Parameters
 FS22SI, HBM	 State Key Laboratory of Fire Science, University of Science and Technology of China	 • Channel: 8 • Measurement time: 1 s • Range: 1500-1600 nm • Resolution: 1 pm • 100001 points per samples
 AQ6380, YOKOGAWA	 Institute of Photonics Technology, Jinan University	 • Channel: 1 • Measurement time: 16 s • Range: 1500-1600 nm • Resolution: 1 pm • 100001 points per samples (HIGH1 mode)

Manuscript modifications: The discussions above have been added in the revised manuscript (see the red colored texts in Methods, page 9) and the supplementary information (see the Table S3, page 20).

[8] References standardization is missing.

Response: We have standardized the references in the revised manuscript.

[9] Similar figures were identified in the Supplementary information document, S6-b (doi.org/10.1016/j.measurement.2022.111961) and S7-c,d,e,f (doi.org/10.1364/OE.22.021757) without any references. It is assumed that one manuscript submitted to a very prestigious journal such as *Nature Communications* should bring excellent and new inputs to the scientific community beyond the state of the art, and this is not verified in this case.

Response: We are sorry for these mistakes! We have modified above two Figures in the Supplementary information document and added two references accordingly [R22, R23].

Figure 12 Femtosecond FBG physical image and microscope image.

Figure 13 Fabrication process of the FPI.

[R22] Liu, Y., Liu, Z., Mei, W., Han, X., Liu, P., Wang, C., Xia, X., Li, K., Wang, S., Wang, Q. & Guo, T. Operando monitoring Lithium-ion battery temperature via implanting femtosecond-laser-inscribed optical fiber sensors. *Measurement* 203, 111961 (2022).

[R23] Wu, C., Liu, Z., Zhang, A. P., Guan, B.-O. & Tam, H.-Y. In-line open-cavity Fabry-Perot interferometer formed by C-shaped fiber for temperature-insensitive refractive index sensing. *Opt. Express* 22, 21757-21766 (2014).

Manuscript modifications: The figures have been added in the revised supplementary information (see the red colored Fig. S7 and Fig. S9, page 9, 11).

[10] *The tube protection diameter used for the fiber optic sensor's instrumentation on the LiB significantly increases their invasiveness. How could this approach impact the results regarding the pressure and temperature flow inside the battery?*

Response: The ceramic tube (made by Al_2O_3) used in this experiment has an outer diameter of 1 mm and an inner diameter of 0.5 mm. As shown in the Figure 14, the FBG temperature sensing area is located outside the casing and placed in the middle of the battery. In addition, there are relevant literatures that use a combination of a ceramic tube with a diameter of 1.55 mm and a thermocouple to measure the internal temperature of the battery [R24]. So the ceramic tube with an outer diameter of 1 mm does not affect the temperature measurement inside the battery.

The FPI pressure sensing area is placed inside the ceramic tube, and the main function of the ceramic tube is to protect the open-cavity FPI from the electrolyte composition inside the battery. As shown in Figure 14, the ceramic tube near the end of the sensor probe is open. According to the principle of the communicating vessels, the pressure sensor placed inside the battery is in a pressure equilibrium state with the internal environment of the battery, so the ceramic tube with an outer diameter of 1 mm does not can affect the pressure measurement inside the battery.

Figure 14 Diagram of assembling the FBG-FPI sensor

[R24] Gulsoy, B., Vincent, T. A., Sansom, J. E. H. & Marco, J. In-situ temperature monitoring of a lithium-ion battery using an embedded thermocouple for smart battery applications. *Journal of Energy Storage* 54, 105260 (2022).

[11] The optical sensing approach used to pressure detection in the manuscript is based on one open Fabry-Perot interferometer. However, in some of the battery applications (EVs) different LiB formats are used. Can you please comment on how this type of sensor can be used for a different LiB configuration, such as the pouch cell or coin cell, and please comment about their battery instrumentation regarding their handling and fragility (due to the C-shape fiber format used)? Is this feasible for a mass manufacturing scale?

Response: 1) For the pouch cell, the open cavity FPI is difficult to insert into the commercial pouch cell because of the absence of hollow part inside the pouch. However, two methods can be taken for internal pressure measurement.

Method 1: The FPI can be used in the modified pouch cell as shown in Figure 15, where a gas bag is usually designed for gas accumulation [R25, R26]. The FPI can be inserted into the gas bag to measure the internal pressure.

Method 2:

1) **For pouch cells**, the surface and internal strain variations can be apparent due to the soft aluminum plastic film, therefore, the internal temperature and strain can be measured by implanting only FBG sensors, avoiding the particularity of C-shaped FPI. This has been implemented in a lithium-sulfur cell to measure the cathode stress evolution [R27]. The internal temperature-strain evolution during thermal runaway of the pouch cell is also worth studying, which may be our future work. We sincerely thank the reviewer for opening up a promising research topic for us!

Figure 15 The FPI insertion into the gas bag of the modified pouch cell [R25, R26]

2) **For coin cells**, the coin cell is usually assembled for new materials development and mechanism elucidation, the internal temperature and pressure change are extremely low attributed to the low capacity. The coin cell can not trigger thermal runaway upon thermal, mechanical and electric abuse. Therefore, it is generally not focused on the temperature and pressure evolution of coin cell.

3) **For large format prismatic cells**, we are now working on the internal temperature-pressure monitoring during thermal runaway for large format prismatic cell (over 200 Ah). The cell is generally composed of two cell units, there is some hollow region between the two cell units which allows the FPI insertion as shown in Figure 16, which also verify the transferability of the methodology to other cell formats.

Figure 16 The transferability of the FBG-FPI methodology to large format cells

4) For the mass manufacturing scale, the open-cavity fibers can be mass-produced through optical fiber drawing towers. The advantage of open-cavity PFI is that the pressure sensitivity does not depend on the cavity length, reducing the number of steps required to calibrate each sensor in practice. And with the development of wavelength division multiplexing, time division multiplexing and frequency division multiplexing technologies, multiplexing of multiple FPI signals in a single fiber can be achieved and is expected to be used in large scale energy storage systems.

To accelerate the application of FPI to energy storage system, the joint efforts of academia and industry are needed. **Firstly**, the sensors need to be improved to enhance the tolerance of the optical fiber sensors to the extreme environment during thermal runaway, and improve their fragility. **Secondly**, it is urgent to cooperate with battery manufacturers to embed optical fibers in the battery manufacturing process to avoid secondary irreversible damage to the battery and ensure signal sensing and battery air tightness.

We have cooperated with **CATL (Contemporary Ampere Technology Co., Limited.)** to produce safe battery implanted with optical fiber sensors, of which the internal temperature and pressure can be monitored simultaneously. We still firmly believe that through the efforts of researchers from academia and industry, optical fiber will definitely be able to be used in energy storage systems for battery safety monitoring in the future!

[R25] Wang, Y., Feng, X., Peng, Y., Zhang, F., Ren, D., Liu, X., Lu, L., Nitta, Y., Wang, L. & Ouyang, M. Reductive gas manipulation at early self-heating stage enables controllable battery thermal failure. *Joule* 6, 2810-2820 (2022).

[R26] Deng, Z., Huang, Z., Shen, Y., Huang, Y., Ding, H., Luscombe, A., Johnson, M., Harlow, J. E., Gauthier, R. & Dahn, J. R. Ultrasonic Scanning to Observe Wetting and "Unwetting" in Li-Ion Pouch Cells. *Joule* 4, 2017-2029 (2020).

[R27] Miao, Z., Li, Y., Xiao, X., Sun, Q., He, B., Chen, X., Liao, Y., Zhang, Y., Yuan, L., Yan, Z., Li, Z. & Huang, Y. Direct optical fiber monitor on stress evolution of the sulfur-based cathodes for lithium-sulfur batteries. *Energ. Environ. Sci.* 15, 2029-2038 (2022).

[12] Almost all the methods reported are ok. However, the authors did not comment on why a mixture of just 3% for each different gas concentration was used (Figure S3). There are already some works reported in the literature (e.g. doi.org/10.1364/OL.37.003063) where it is shown that the sensitivity of open FPI sensors changes in presence of different gases (due to refractive index variations, which are different for each gas).

Response: As described in the literature [R28], the sensitivity of open FPI sensors changes with the presence of different gases due to the different refractive index for each gas. It can be seen from the literature [R29], for 18650 LiFePO₄ batteries, the gases produced during thermal runaway are mainly CO₂, H₂ and some hydrocarbons. Due to the flammable nature of hydrogen (the explosion limit is 4%-75.6% volume concentration), laboratory measurements can only be achieved for hydrogen mixtures at 3% concentration (We can only purchase commercial hydrogen mixtures at 3% concentration for safety). Therefore, a uniform 3% gas mixture was chosen for comparison purposes. The FBG-FPI sensor was found to exhibit excellent linear responsiveness and measurement accuracy (pressure sensitivity error of approximately 0.5% for different gas compositions).

[R28] Coelho, L., Tafulo, P. A. R., Jorge, P. A. S., Santos, J. L., Viegas, D., Schuster, K., Kobelke, J. & Frazao, O. Simultaneous measurement of partial pressure of O₂ and CO₂ with a hybrid interferometer. *Opt. Lett.* 37, 3063-3065 (2012).

[R29] Golubkov, A. W., Fuchs, D., Wagner, J., Wiltse, H., Stangl, C., Fauler, G., Voitic, G., Thaler, A. & Hacker, V. Thermal-runaway experiments on consumer Li-ion batteries with metal-oxide and olivin-type cathodes. *RSC Advances* 4, 3633-3642 (2014).

[13] Also, the deduction performed from equation 4 does not seem to be correct. There are several studies in the literature regarding the pressure variation with open cavities in air related to the refractive index shift. The n_m parameter is not equal to P values. Therefore, the P value in Equation 5 is not correct. The authors are advised to check the following references and correct this Section.

doi.org/10.1109/JPHOT.2015.2489926;

doi.org/10.1364/OE.23.023484.

Response: Thank you for your careful reminder of this manuscript, we have checked the relevant references you provided and corrected this section in the manuscript.

The wavelength spacing between two adjacent dips of the reflected spectrum is known as the free spectral range (FSR).

$$FSR_{FP} = \frac{\lambda_1 \lambda_2}{2n_m L} \quad (3)$$

where λ_1 and λ_2 are wavelengths of two adjacent dips in the FPI reflection spectrum.

For the FPI pressure sensor with the open cavity structure, the gas refractive index change caused by the pressure change is calculated according to Eq. (4) [R30, R31]

$$n_m = 1 + \frac{2.8793 \times 10^{-9} \times P}{1 + 0.003661 \times T} \quad (4)$$

Where P and T represent pressure (Pa) and temperature (°C), respectively.

When FPI is used for pressure measurements, the dip wavelength λ_{FP} of the interferometric spectrum varies with pressure. From this, we obtain the gas pressure sensitivity as:

$$\frac{d\lambda_{FP}}{dP} = \lambda_{FP} \left(\frac{1}{L} \frac{dL}{dP} + \frac{1}{n_m} \frac{dn_m}{dP} \right) \quad (5)$$

where $\frac{1}{L} \frac{dL}{dP}$ represents the change in cavity length with pressure and $\frac{1}{n_m} \frac{dn_m}{dP}$ represents the change in refractive index of the gas with pressure.

The cavity length L of the PFI can be regarded as constant when the pressure changes. Therefore, Eq. (5) can be simplified as follows:

$$\frac{d\lambda_{FP}}{dP} = \frac{\lambda_{FP}}{n_m} \frac{dn_m}{dP} \quad (6)$$

where $\frac{dn_m}{dP}$ is the rate of change of the refractive index of the gas with pressure.

Eq. (6) shows that the wavelength shift of the interference spectrum is linearly related to the change in the refractive index of the gas in the cavity caused by the change in pressure. Thus, the pressure change inside the cell is monitored by the shift of the FPI peak wavelength.

[R30] Tang, J., Yin, G., Liao, C., Liu, S., Li, Z., Zhong, X., Wang, Q., Zhao, J., Yang, K. & Wang, Y. High-Sensitivity Gas Pressure Sensor Based on Fabry–Pérot Interferometer With a Side-Opened Channel in Hollow-Core Photonic Bandgap Fiber. *IEEE Photonics Journal* 7, 1-7 (2015).

[R31] Xu, B., Wang, C., Wang, D. N., Liu, Y. & Li, Y. Fiber-tip gas pressure sensor based on dual capillaries. *Opt. Express* 23, 23484-23492 (2015).

Manuscript modifications: The discussions above have been added in the revised manuscript (see the red colored texts in Methods, page 9-10 and Refs. 35, 36).

[14] *The equation used to describe the FPI interference condition is not correct (please see page 3, 1st paragraph, and in Figure 1f). As it is, the equation defines the free spectral range.*

Response: Thank you for your careful reminder of this manuscript, we have checked the relevant references you provided and corrected this section in the manuscript.

According to the phase matching condition of the FPI, when the initial phase of the interference $\varphi_0 = 0$, if we track the k^{th} order spectral dip wavelength (λ_{FP}) of the FPI, its phase $4\pi n_m L / \lambda_{FP}$ keeps a constant value of $(2k + 1)\pi$ [R32]. Thus we have

$$4\pi n_m L / \lambda_{FP} = (2k + 1)\pi \quad (7)$$

Eq. (7) can be simplified to obtain:

$$\lambda_{FP} = 4n_m L / (2k + 1) \quad (8)$$

where k is an integer, L is the distance between the two reflection planes and n_m is the refractive index of the medium in between the planes.

[R32] Wei, T., Han, Y., Li, Y., Tsai, H.-L. & Xiao, H. Temperature-insensitive miniaturized fiber inline Fabry-Perot interferometer for highly sensitive refractive index measurement. *Opt. Express* 16, 5764-5769 (2008)

Manuscript modifications: The discussions above have been added in the revised manuscript (see the red colored texts in Fig. 1i on page 2, Equations on page 3 and Refs. 33).

[15] During the fabrication of the FBG-FPI, the authors did not clarify which part of the sensor is created first. On page 9, the authors mention that “The cut open - cavity fiber is then spliced to another SMF using manual alignment, and the other end is pinched off directly to avoid unwanted reflections”. However, the FBG sensors seem to be created after the FPI sensors. What is the distance between the FPI and fiber end? What is the impact of having a low reflectivity FPI before/after the FBG sensor (in relation to the interrogation system)? More details should be provided.

Response: Fabrication of the FBG-FPI: First the open cavity FPI part of the sensor is created and then the FBG is inscribed by a femtosecond laser point by point at a distance of 10 mm from the FPI as shown in Figure 17. The length of the FBG is 0.8 mm, so the distance between the FPI and the optical fiber end is approximately 12 mm. We have redrawn the Fig.1 to display the geometrical depiction and distance between FBG and FPI.

Figure 17 Fabrication process of the FBG-FPI sensor

The impact of having a low reflectivity FPI before/after the FBG sensor can depend on the specific system configuration and the intended application.

If a low reflectivity FPI is placed before the FBG sensor in a system, it can act as a filter, reducing the amount of extraneous light that reaches the FBG. This can be useful for improving the signal-to-noise ratio and reducing the impact of environmental noise sources like temperature fluctuations or mechanical vibrations.

If a low reflectivity FPI is placed after the FBG sensor in a system, it can act as a tunable filter, allowing specific wavelengths of light to be transmitted or blocked. This can be useful for improving the selectivity and specificity of the system, especially if the FBG is used to detect a specific wavelength shift caused by a particular phenomenon like strain or temperature changes.

Therefore, the specific reflectivity values that work best will depend on the specific characteristics of the system, including the light source, detection system, and optical fiber properties.

Manuscript modifications: The discussions above have been added in the revised manuscript (see the red colored texts in Fig.1, page 2 and Methods, page 9) and revised supplementary information (see the red colored Fig. S10, page 11).

[16] In the FPI production, what is the uncertainty associated with the fabrication method?

Response: The uncertainty associated with FPI fabrication is mainly due to the difference in cavity length. As shown in the Figure 18 (Fig. S8 in Supplementary Information), we investigate the effect of different cavity lengths of FPI on the pressure sensitivity, and the experiments show that the cavity length only affects the intensity and free spectral range (FSR) of the FPI spectrum, and the response sensitivity to pressure is basically the same (4185 pm MPa⁻¹ for the reflection peak wavelength near 1550 nm, linearity R² = 99.9%).

Figure 18 The FPI's sensing characteristics with different cavity lengths

Thus, the manuscript in its present state and whit all the early points mentioned does not present the novelty and quality for being published in the prestigious Nature Communications Journal. It is my opinion that the manuscript should be rejected for publication in this Journal.

Response: We appreciate the reviewer's critical judgement. Your constructive comments do help in improving and strengthening our paper. The key point for this work is it focused on battery thermal runaway and will be significant in the field of improved battery management systems (as we have carefully discussed and answered in your first important question, please see the two advantage points and Figure 7 and 8 in pages of 11 and 12).

Having done our best to satisfactorily answer the referee comments and requests for extra data, we hope that this revised version could meet the high standards of *Nature Communications*.

REVIEWERS' COMMENTS

Reviewer #2 (Remarks to the Author):

Dear authors,

Thank you for revising and performing significant corrections and improvements in the manuscript in order to increase its quality and significance. I am also satisfied with most responses given. However, in the introduction section, the following sentences should be rewritten, where there are some important references missing: "Subsequently, they further proposed advanced sensing schemes for cells internal strain and temperature discrimination by using a polarization-maintaining FBG [19] and a Fabry- Perot interferometer (FPI) [20]. This was followed by multi-point measurements in another group [21]". Reading the works that you reference here, the reference [20] used an optical fiber hybrid sensors configuration based on FBG 's and Fabry-Perot interferometers for simultaneous temperature and strain discrimination (a similar sensing approach that is presented in this manuscript), not only a Fabry-Perot interferometer as you refer to in the added sentence. Also, several studies about multi-point measurements in LiB surfaces were reported by using FBG sensors and thermocouples (by Pinto 's group) before the study referenced in [21]:

[1] doi.org/10.3390/batteries4040067;

[2] doi.org/10.1016/j.applthermaleng.2018.12.135;

[3] doi.org/10.1016/j.measurement.2017.07.049.

I would suggest adding the reference [R23] to the Supplementary Information and citing them in Figure 1 g, and in the Caption of Fig. S7, as the photos are still the same as in the above-mentioned reference.

Therefore, I would like to claim that the manuscript in its current form should be minor reviewed before publication in Nature Communications.

Manuscript ID: NCOMMS-23-03412A

Title: Operando monitoring of thermal runaway in commercial lithium-ion cells via advanced lab-on-fiber technologies

Reviewer #2:

Dear authors,

Thank you for revising and performing significant corrections and improvements in the manuscript in order to increase its quality and significance. I am also satisfied with most responses given. However, in the introduction section, the following sentences should be rewritten, where there are some important references missing: "Subsequently, they further proposed advanced sensing schemes for cells internal strain and temperature discrimination by using a polarization-maintaining FBG [19] and a Fabry–Perot interferometer (FPI) [20]. This was followed by multi-point measurements in another group [21]". Reading the works that you reference here, the reference [20] used an optical fiber hybrid sensors configuration based on FBG's and Fabry-Perot interferometers for simultaneous temperature and strain discrimination (a similar sensing approach that is presented in this manuscript), not only a Fabry-Perot interferometer as you refer to in the added sentence. Also, several studies about multi-point measurements in LiB surfaces were reported by using FBG sensors and thermocouples (by Pinto's group) before the study referenced in [21]:

[1] doi.org/10.3390/batteries4040067;

[2] doi.org/10.1016/j.applthermaleng.2018.12.135;

[3] doi.org/10.1016/j.measurement.2017.07.049.

I would suggest adding the reference [R23] to the Supplementary Information and citing them in Figure 1 g, and in the Caption of Fig. S7, as the photos are still the same as in the above-mentioned reference.

Therefore, I would like to claim that the manuscript in its current form should be minor reviewed before publication in Nature Communications.

Response: We appreciate your so careful review and helpful suggestions.

(1) We have rewritten the text in the Introduction to highlight the pioneering contributions of Pinto's group and the following 3 papers in the References.

(see revised manuscript, the red colored texts in Introduction, page 1, Refs. 19, 20, 21)

"Integrating optical fiber sensors inside batteries was first reported by Pinto's group for real-time temperature measurements¹⁸, following with a series of studies for multi-point measurements¹⁹⁻²². Subsequently, they further proposed advanced sensing schemes for cells internal strain and temperature discrimination by using a polarization-maintaining FBG²³ and a hybrid FBG and Fabry–Perot interferometer (FPI)²⁴."

[19] Nascimento, M., Ferreira, M. S. & Pinto, J. L. Real time thermal monitoring of lithium batteries

with fiber sensors and thermocouples: A comparative study. Measurement 111, 260-263 (2017).
[20] Nascimento, M., Paixão, T., Ferreira, M. S. & Pinto, J. L. *Thermal Mapping of a Lithium Polymer Batteries Pack with FBGs Network. Batteries 4, 67 (2018).*
[21] Nascimento, M., Ferreira, M. S. & Pinto, J. L. *Temperature fiber sensing of Li-ion batteries under different environmental and operating conditions. Applied Thermal Engineering 149, 1236-1243 (2019).*

(2) We have added the reference (Wu, C., Liu, Z., Zhang, A. P., Guan, B.-O. & Tam, H.-Y. In-line open-cavity Fabry-Perot interferometer formed by C-shaped fiber for temperature-insensitive refractive index sensing. *Opt. Express* 22, 21757-21766 (2014)) to the Supplementary Information. And this paper has also been cited in Figure 1g, and in the Caption of Fig. S7. (See revised supplementary information, the red colored Fig. S7, page 9, Ref. 5).